# anTraX, a software package for high-throughput video tracking of color-tagged insects

Asaf Gal*, Jonathan Saragosti, Daniel JC Kronauer*

Laboratory of Social Evolution and Behavior, The Rockefeller University, New York, United States

**Abstract** Recent years have seen a surge in methods to track and analyze animal behavior. Nevertheless, tracking individuals in closely interacting, group-living organisms remains a challenge. Here, we present anTraX, an algorithm and software package for high-throughput video tracking of color-tagged insects. anTraX combines neural network classification of animals with a novel approach for representing tracking data as a graph, enabling individual tracking even in cases where it is difficult to segment animals from one another, or where tags are obscured. The use of color tags, a well-established and robust method for marking individual insects in groups, relaxes requirements for image size and quality, and makes the software broadly applicable. anTraX is readily integrated into existing tools and methods for automated image analysis of behavior to further augment its output. anTraX can handle large-scale experiments with minimal human involvement, allowing researchers to simultaneously monitor many social groups over long time periods.

## Introduction

Our understanding of behavior, together with the biological, neural, and computational principles underlying it, has advanced dramatically over recent decades. Consequently, the behavioral and neural sciences have moved to study more complex forms of behavior at ever-increasing resolution. This has created a growing demand for methods to measure and quantify behavior, which has been met with a wide range of tools to measure, track, and analyze behavior across a variety of species, conditions, and spatiotemporal scales (*Anderson and Perona, 2014*; *Berman, 2018*; *Brown and de Bivort, 2018*; *Krakauer et al., 2017*; *Dell et al., 2014*; *Robie et al., 2017a*; *Todd et al., 2017*; *Egnor and Branson, 2016*). One of the exciting frontiers of the field is the study of collective behavior in group-living organisms and particularly the behavior of groups of insects. Insects provide an attractive and powerful model for collective and social behavior, as they exhibit a wide range in social complexity, from solitary to eusocial, while allowing for controlled, high-throughput experiments in laboratory settings (*Feinerman and Korman, 2017*; *Lihoreau et al., 2012*; *Gordon, 2014*; *Schneider et al., 2012*). However, although complex social behavior has been the focus of extensive research for over a century, technological advances are only beginning to enable systematic and simultaneous measurements of behavior in large groups of interacting individuals.

Solutions for automated video tracking of insects in social groups can be roughly divided into two categories (for reviews see *Dell et al., 2014*; *Robie et al., 2017a*): methods for tracking unmarked individuals (*Branson et al., 2009*; *Pérez-Escudero et al., 2014*; *Romero-Ferrero et al., 2019*; *Sridhar et al., 2019*; *Feldman et al., 2012*; *Khan et al., 2005*; *Fasciano et al., 2014*; *Fasciano et al., 2013*; *Bozek et al., 2020*), and methods for tracking marked individuals (*Mersch et al., 2013*; *Robinson et al., 2012*). The former category has the obvious advantages of reduced interference with natural behavior, unbounded number of tracked individuals, and not

*For correspondence:
asafg1@gmail.com (AG);
dkronauer@rockefeller.edu (DJCK)

**Competing interests:** The authors declare that no competing interests exist.

having the burden of tagging animals and maintaining these tags throughout the experiment. At the same time, these approaches, when applied to individual tracking, are limited by a more extensive computational burden, higher error rates, and stricter requirements for image quality. The most common approach for tracking unmarked individuals is to try and follow the trajectory of an individual for the duration of the video. The challenge in this approach is to resolve individuals from each other and link their locations in consecutive frames during close range interactions, when they are touching or occluding each other. Common solutions to this problem are to employ sophisticated segmentation methods (*Branson et al., 2009*; *Pérez-Escudero et al., 2014*; *Sridhar et al., 2019*), to use predictive modeling of the animals' motion (*Branson et al., 2009*; *Fasciano et al., 2013*), or to use image characteristics to match individuals before and after occlusions (*Fasciano et al., 2014*). The success of these solutions is case-specific and will usually be limited to relatively simple problems, where interactions are brief, occlusion is minimal, or image resolution is sufficient to resolve the individuals even during an interaction. One important limitation of this approach is that no matter how low the error rate is, it tends to increase rapidly with the duration of the experiment. The reason is that once identities are swapped, the error is unrecoverable, and will propagate from that moment on. A different algorithmic approach for tracking unmarked individuals is to use object recognition techniques to assign separate pieces of trajectories to the same individual (*Pérez-Escudero et al., 2014*; *Romero-Ferrero et al., 2019*). While this approach is promising and performs well on many tracking problems, it requires high image quality to identify unique features for each individual animal. It will also generally not perform well on animals with high postural variability and is hard to validate on large datasets.

On the other hand, tagging individuals with unique IDs has the advantage of having a stable reference, enabling error recovery. This approach also provides a simpler method for human validation or correction and enables following the behavior of individuals even if they leave the tracked region, or across experiments when the same animals are tested in different conditions. The use of general-purpose libraries such as AprilTags (*Mersch et al., 2013*; *Olson, 2011*; *Heyman et al., 2017*; *Greenwald et al., 2018*; *Stroeymeyt et al., 2018*) and ArUco (*Garrido-Jurado et al., 2014*), or application-specific patterned tags (*Crall et al., 2015*; *Boenisch et al., 2018*; *Wario et al., 2015*; *Wild et al., 2018*), has become the gold standard for this approach in recent years. However, these tags are applicable only to species with body sizes sufficiently large to attach them, have adverse effects on the animals' behavior, and are often lost during experiments. They also require relatively high image resolution to correctly read the barcode pattern. Taken together, while currently available methods cover a wide range of experimental scenarios, the ability to accurately track the behavior of animals in groups remains one of the major hurdles in the field. As a result, much of the experimental work still relies on manual annotation, or on partially automated analysis pipelines that require considerable manual effort to correct computer-generated annotations (see *Aguilar et al., 2018*; *Gelblum et al., 2015*; *Leitner and Dornhaus, 2019*; *Valentini et al., 2020* for recent examples). In principle, marked animals can also be tracked by general-purpose image-based trackers such as idTracker.ai, supplementing the pixel information of the animals' appearances with artificial features. To the best of our knowledge, however, this approach has not been formally described, and it can be expected to perform less well than trackers specifically designed for a given marking technique.

Here, we present anTraX, a new software solution for tracking color-tagged insects and other small animals. Color tagging is one of the best-established and widely used methods to mark insects, both in the field and in the laboratory (*Leitner and Dornhaus, 2019*; *Valentini et al., 2020*; *Walker and Wineriter, 1981*; *Gordon, 1989*; *Hagler and Jackson, 2001*; *Ulrich et al., 2018*; *Holbrook et al., 2011*; *Holbrook, 2009*), with long-term durability and minimal effects on behavior. anTraX works by combining traditional segmentation-based object tracking with image-based classification using convolutional neural networks (CNNs). In addition, anTraX uses a graph object for representing tracking data (*Nillius et al., 2006*), enabling the inference of identity of unidentified objects by propagating temporal and spatial information, thereby optimizing the use of partial tag information. anTraX is uniquely suited for tracking small social insects that form dense aggregates, in which individuals are unidentifiable over large parts of the experiment even for the human observer. It will also be useful in tracking and analyzing behavior in heterogenic groups of 'solitary' insects, where keeping track of the individual identity for long experimental durations is important. Such experiments are of increasing interest, as the study of behavior in classical model systems like

*Drosophila* fruit flies is shifting toward understanding more complex behavioral phenomena such as social interactions, individuality and inter-species interactions (*Schneider et al., 2012*; *Seeholzer et al., 2018*; *Schneider and Levine, 2014*; *Honegger and de Bivort, 2018*; *Ayroles et al., 2015*; *Akhund-Zade et al., 2019*).

While we tested anTraX and found it useful for behavioral analyses in a range of study systems, it was specifically developed for experiments with the clonal raider ant *Ooceraea biroi*. The clonal raider ant is an emerging social insect model system with a range of genomic and functional genetic resources (*Ulrich et al., 2018*; *Oxley et al., 2014*; *Trible et al., 2017*; *McKenzie and Kronauer, 2018*; *Chandra et al., 2018*; *Teseo et al., 2014*). The unique biological features of the species enable precise control over the size, demographics and genetic composition of the colony, parameters that are essential for systematic study of collective behavior in ants (*Ulrich et al., 2018*; *Chandra et al., 2020*). Moreover, the species is amenable to genetic manipulations (*Trible et al., 2017*), which opens new possibilities not only for understanding the genetic and neural bases of social and collective behaviors, but also for developing and validating theoretical models by manipulating behavioral rules at the level of the individual and studying the effects on group behavior. While these ants have great promise for the study of collective behavior, they are hard to track using available approaches, due to their small size and tendency to form dense aggregates. anTraX thus constitutes a crucial element in the clonal raider ant toolbox, enabling researchers to characterize behavior in unprecedented detail both at the individual and collective level.

anTraX was designed with large-scale behavioral experiments in mind, where hundreds of colonies are recorded in parallel for periods of weeks or months, making manual tracking or even error correction impractical. Its output data can be directly imported into software packages for higher level analysis of behavior (e.g. *Kabra et al., 2013*) or higher resolution postural analysis of individuals in the group (*Pereira et al., 2019*; *Mathis et al., 2018*; *Berman et al., 2014*; *Graving et al., 2019*). This enables the utilization of these powerful tools and methods for the study of social insects and collective behavior. anTraX is modular and flexible, and its many parameters can be set via a graphical interface. The software is open source, and its main algorithmic components can be easily modified. Here, we provide a brief description of the different steps and principles that constitute the anTraX algorithm, while a full description is given in the Appendix and the online documentation. We validate the performance of anTraX using a number of benchmark datasets that represent a variety of behavioral settings and experimental conditions.

## Materials and methods

The anTraX algorithm consists of three main steps. First, similar to other multi-object tracking algorithms (*Pérez-Escudero et al., 2014*; *Romero-Ferrero et al., 2019*), it segments the frames into background and ant-containing *blobs* and organizes the extracted blobs into trajectory fragments termed *tracklets*. The tracklets are linked together to form a directed *tracklet graph* (*Nillius et al., 2006*). The second step of the algorithm is tracklet classification, in which identifiable single-animal tracklets are labeled with a specific ID by a pre-trained CNN, while other tracklets are marked as either unidentified single-animal tracklets, or as multi-animal tracklets. Third, we infer the identity of unclassifiable tracklets in the tracklet graph by using temporal, spatial and topological propagation rules.

### Object tracking and construction of the tracklet graph

Each frame is subtracted from the background, and a fixed threshold is applied to segment the frame into background regions and animal-containing blobs to be tracked. When two or more animals are close together, they will often be merged into a single larger blob (*Figure 1A–C*). Unlike other tracking solutions, we do not attempt to separate these larger blobs into single animal blobs at this stage, because those attempts are based on heuristic decisions that do not generalize well across species and experimental conditions. Instead, we will infer the composition of these larger blobs from the tracklet graph in a later step. Each blob in the currently processed frame is then linked to blobs in the previous frame (*Figure 1D–E*). A link between a blob in frame *t* and a blob in frame *t-1* implies that some or all of the animals that were part of the first blob, are present in the second one. A blob can be linked to one blob (the simplest case, where the two blobs have the same ant composition), to a few blobs (where animals leave or join the blob), or to none (suggesting

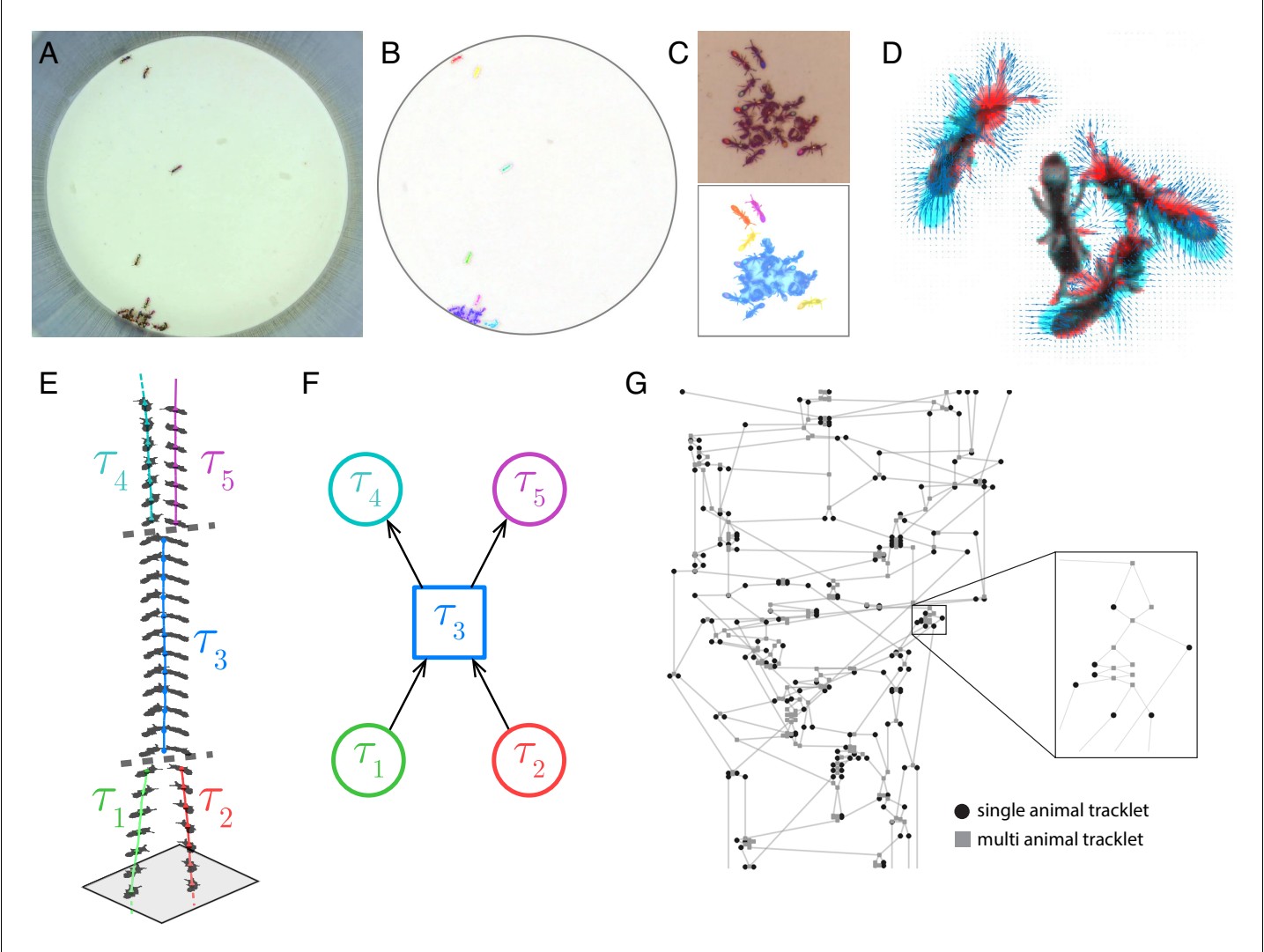

**Figure 1.** Blob tracking and construction of the tracklet graph. (**A**) An example frame from an experiment with 16 ants marked with two color tags each. (**B**) The segmented frame after background subtraction. Each blob is marked with a unique color. Some blobs contain single ants, while others contain multiple ants. (**C**) A higher resolution segmentation example. While some ants are not distinguishable from their neighbors even for the human eye, others might be segmented by tuning the segmentation parameters, or by using other, more sophisticated segmentation algorithms. The anTraX algorithm takes a conservative approach and leaves those cases unsegmented to avoid segmentation errors. (**D**) Optical flow is used to estimate the 'flow' of pixels from one frame to the next, giving an approximation of the movements of the ants. The cyan silhouettes represent the location of an ant in the first frame, and the red silhouettes represent the location in the second frame. The results of the optical flow procedure are shown with blue arrows, depicting the displacement of pixels in the image. (**E**) An example of constructing and linking tracklets. Each layer represents a section of segmented frame. Two ants are approaching each other (tracklets marked $\tau_1$ and $\tau_2$), until they are segmented together. At that point, the two tracklets end, and a third multi-ant tracklet begins ($\tau_3$). Once the two ants are again segmented individually, the multi-ant tracklet ends, and two new single-ant tracklets begin ($\tau_4$ and $\tau_5$). (**F**) The graph representation of the tracklet example in **E**. (**G**) A tracklet graph from an experiment with 36 ants, representing 3 min of tracking data. The nodes are located according to the tracklet start time on the vertical axis, beginning at the bottom. The inset depicts a zoomed-in piece of the graph.

the animals in the blob were not detected in the other frame). We use Optical Flow to decide which blobs should be linked across frames (*Figure 1D*). While Optical Flow is computationally expensive, we found it to be significantly more accurate than alternatives such as simple overlap or linear assignment (based either on simple spatial distance or on distance in some feature space). To reduce the computation cost, we run the optical flow in small regions of the image that contain more than one linking option (see Appendix section 1.4 for details).

Blobs are organized into tracklets, defined as a list of linked blobs in consecutive frames that are composed of the same group of individuals (*Figure 1E–F*). Following linkage, tracklets are updated in the following way: (i) A blob in the current frame *t* that is not linked to any blob in the previous frame *t-1*, will 'open' a new tracklet. (ii) A blob in the previous frame that is not linked to any blob in the current frame, will 'close' its tracklet. (iii) If a pair of blobs in the previous and current frames are exclusively linked, the current blob will be added to the tracklet that contains the previous blob. (iv) Whenever a blob in the current or previous frames is connected to more than one blob, the tracklets of the linked blobs in the previous frames will 'close', and new tracklets will 'open' with the blobs in the current frame. In these latter cases, the linking between the blobs across different tracklets will be registered as an edge in the directed tracklet graph from the earlier tracklet to the latter. The tracklet graph is constructed by running an iterative loop over all the frames in the experiment. The result of this part of the algorithm, after processing all frames in the video, is a directed acyclic graph containing references to all tracklets and blobs in the dataset (*Figure 1G*).

## Tracklet classification

The next step is tracklet classification, in which we label tracklets containing single animals that can be reliably identified with a specific ID (Appendix section 2.3). The successful propagation of individual IDs on top of the tracklet graph requires at least one identification of each ID at this step. Propagation will improve with additional independent identifications of individuals throughout the video. Nevertheless, it is important to note that our approach does not rely on the identification of each and every tracklet, but rather on inferring the composition of tracklets based on propagation of IDs on top of the tracklet graph. Hence, we apply a conservative algorithm that classifies only reliable cases and leaves ambiguous ones as unidentified. Classification is done by training and applying a convolutional neural network (CNN) on each blob image in the tracklet. The most frequent ID is then applied to the entire tracklet (*Figure 2A*). In addition to the ID label, we also assign a classification confidence score to each classified tracklet, which takes into account the number of identified blobs in the tracklet, the confidence of each classification, and the prevalence of contradictory classifications across blobs in the tracklet (see Appendix section 2.4). anTraX comes with a graphical interface for training, validating, and running the CNN (see Supplementary Material and online documentation).

## ID propagation

The last part of the algorithm is the propagation of ID assignments on the tracklet graph. While formal approaches for solving this problem using Bayesian inference have been proposed (*Nillius et al., 2006*), we chose to implement an ad-hoc greedy iterative process that we found to work best in our particular context. Each node in the graph (corresponding to a tracklet) is annotated with a dynamic list of assigned IDs (IDs that *are* assigned to the tracklet) and a list of possible IDs (IDs that *might* be assigned to the tracklet, i.e., that were not yet excluded). Initially, all nodes are marked as 'possible' for all IDs, and no IDs are assigned to any nodes. All the classified tracklets from the previous step are now ranked by their confidence score. Starting with the highest confidence tracklet, its ID is propagated on the graph as far as possible. Propagation is done vertically on the graph on top of edges, both positively (an ID that is assigned to a node must also be assigned to at least one of its successors and one of its predecessors) and negatively (an ID cannot be in the possible list of a node, if it is not in at least one successor and one predecessor node), horizontally (if an ID is assigned to a node, it cannot be assigned to any other time-overlapping node), and using topological constraints (*Figure 2B*, *Figure 2—video 1*). Only non-ambiguous propagation is performed, and propagation is halted whenever an ambiguity or contradiction arises. We iterate the propagation until no more assignments can be made. Some of the propagation rules are modified in cases of tracklets that start or end in regions where individuals can enter or leave the tracked area (see Appendix section 3). *Figure 2C–F* visualizes an example of tracking an ant throughout a 10 min segment from an actual experiment and depicts the path of the ant through the tracklet graph along with its spatial trajectory.

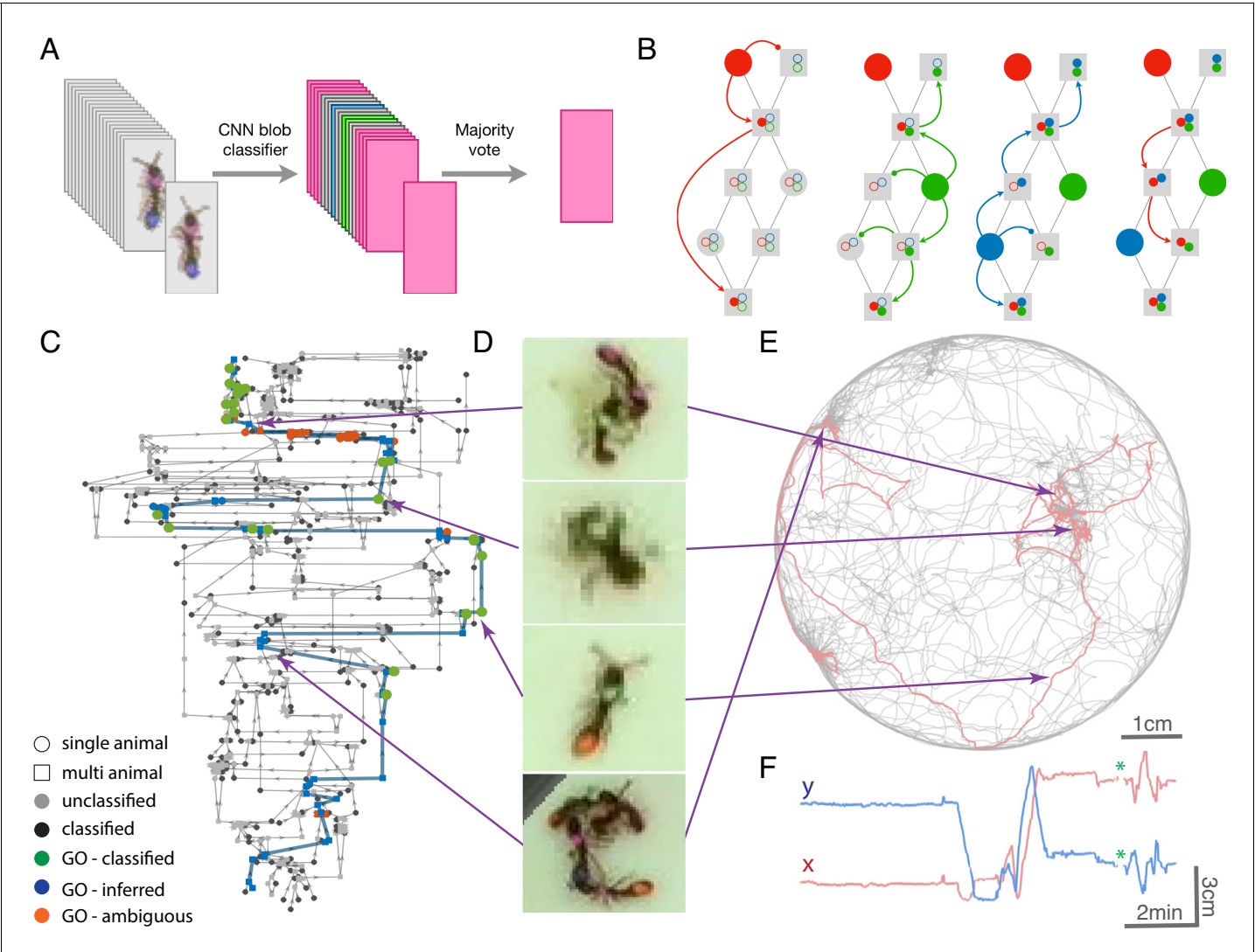

**Figure 2.** Tracklet classification and ID propagation on the tracklet graph. (**A**) Schematic of the tracklet classification procedure. All blobs belonging to the tracklet are classified by a pre-trained CNN classifier. The classifier assigns a label to each blob, which can be an individual ID (depicted as colored rectangles in the figure), or an ambiguous label ('unknown', depicted in gray). The tracklet is then classified as the most abundant ID in the label set, along with a confidence score that depends on the combination of blob classifications and their scores (see Supplementary Material for details). (**B**) A simple example of propagating IDs on top of the tracklet graph. The graph represents a tracking problem with three IDs (represented as red/blue/green) and eight tracklets, of which some are single-animal (depicted as circles) and some are multi-animal (depicted as squares). Three of the single-animal tracklets have classifications, and are depicted as color-filled circles. The graph shows how, within four propagation rounds, assigned IDs are propagated as far as possible, both negatively (round head arcs) and positively (arrow heads), until the animal composition of all nodes is fully resolved. See also *Figure 2—video 1* for an expanded animated example. (**C**) An example of a solved tracklet graph from an experiment with 16 ants, representing 10 min of tracking. Single ant tracklets are depicted as circle nodes and multi ant tracklets are depicted as square nodes. Black circles represent single ant tracklets that were assigned an ID by the classifier. A subgraph that corresponds to a single focal ant ID ('GO': an ant marked with a green thorax tag and an orange abdomen tag) is highlighted in color. Green nodes represent single ant tracklets assigned by the classifier. Blue nodes represent tracklets assigned by the propagation algorithm. Red nodes are residual ambiguities. (**D**) Example snapshots of the focal ant GO at various points along its trajectory, where it is often unidentifiable. The second image from the bottom shows an image where the ant is identifiable. While the third image from the bottom shows an unidentifiable ant, it belongs to a tracklet which was assigned an ID by the classifier based on other frames in the tracklet. The first and last images show the focal ant inside aggregations, and were assigned by the propagation algorithm. The purple arrows connect each image to its corresponding node in C. (**E**) The 10-min long trajectories corresponding to the graph in C. The trajectory of the focal ant GO is plotted in orange, while the trajectories of all other ants are plotted in gray. Purple arrows again point from the images in D to their respective location in the trajectory plot. (**F**) Plot of the x and y coordinates of the focal ant during the 10 min represented in the graph in C. Gaps in the plot (marked with green asterisks) correspond to ambiguous segments, where the algorithm could not safely assign the ant to a tracklet. In most cases, these are short gaps when the ant does not move, and they can be safely interpolated to obtain a continuous trajectory.

*Figure 2 continued on next page*

*Figure 2 continued*

The online version of this article includes the following video for figure 2:

**Figure 2—video 1.** An animated example of the graph propagation algorithm.

https://elifesciences.org/articles/58145#fig2video1

### Export positional and postural results for analysis

The tracking results are saved to disk and can be accessed using supplied MATLAB and Python interface functions. For each individual ID in the experiment, a table is returned, containing its assigned spatial coordinates in each frame of the experiment, and a flag indicating the type of the location estimation (e.g. direct single-animal classification, inferred single-animal, multi-animal tracklet). For frames where the location is derived from single animal tracklets (i.e. the animal was segmented individually), the animal orientation is also returned. Locations estimated from multi-animal tracklets are necessarily less accurate than locations from single-animal tracklets, and users should be aware of this when analyzing the data. For example, calculating velocities and orientations is only meaningful for single-animal tracklet data, while spatial fidelity can be estimated based also on approximate locations. A full description of how to import and process the tracking results is provided in Appendix section 3.6 and the online documentation.

### User interface and parameter tuning

anTraX has many parameters that control the image segmentation step, the classifier architecture and training procedure, and the propagation algorithm. The optimal value for each depends on the specific nature and settings of the processed experiment, from the resolution and quality of the camera, to the details of the organisms and number of tags. anTraX comes with a graphical user interface to tune and verify the value of these parameters. anTraX also contains a user interface for creating an image database and training the CNN for tracklet classification.

### Parallelization and usage on computer clusters

anTraX was specifically designed to process large-scale behavioral experiments, which can contain hundreds of video files and tens of terabytes of data. anTraX includes scripts to process such large datasets in batch mode where individual video files are tracked in parallel on multicore computers and high-performance computer clusters. Following per-video processing, anTraX will run a routine to 'stitch' the results of the individual files together (see online documentation).

### Availability and dependencies

The core tracking steps of anTraX are implemented using MATLAB version 2019a, while the classification parts are implemented using TensorFlow v1.15 in the Python 3.6 environment. Compiled binaries are available for use with the freely available MATLAB Runtime Library and can be run with a command line interface. anTraX can be run on Linux/OSX systems, and large datasets benefit considerably from parallelization on computer clusters. anTraX depends on the free FFmpeg library for handling video files. The result files are readable with any programming language, and we supply a Python module for easily interfacing with output data. anTraX is distributed under the GPLv3 license, and its source code and binaries are freely available (*Gal et al., 2020a*). anTraX is a work in progress and will be continuously extended with new features and capabilities. Online documentation for installing and using the software is available at http://antrax.readthedocs.io. Users are welcome to subscribe, report issues, and suggest improvements using the GitHub interface.

## Results

### anTraX tracks individual ants with near-human accuracy over a wide range of conditions

As any tracking algorithm, the performance of anTraX depends on many external factors, such as the image quality, the framerate, the quality of the color tags (size, color set, number of tags per individual), and the behavior of the organisms (e.g. their tendency to aggregate, their activity level,

etc). anTraX was benchmarked using a number of datasets spanning a variety of experimental conditions (e.g. image quality and resolution, number of tracked individuals, number of tags and colors, size variability in the colony) and study organisms, including four different ant species, as well as the fruit fly *Drosophila melanogaster* (*Table 1*, *Figure 3* and its supplements). All benchmark datasets, together with the raw videos, full description, configuration files, and trained classifiers are available for download (*Gal et al., 2020b*).

The performance of the tracking algorithm can be captured using two separate measures. The first is the rate of assignment, defined as the ratio of assigned locations in the experiments to the total possible assignments (i.e. the number of IDs times the number of frames). The second measure is the assignment error, defined as the ratio of wrong assignments to the total number of assignments made. While the assignment rate can be computed directly and precisely from the tracking results, the error rate in assigning IDs for a given data set needs to be tested against human annotation of the same dataset. Because the recording duration of these datasets is typically long (many hours), it is impractical to manually annotate them in full. Instead of using fewer or smaller datasets, which would have introduced a sampling bias, we employed a validation approach in which datasets were subsampled in a random and uniform way. In this procedure, a human observer was presented with a sequence of randomly selected test points, where each test point corresponded to a location assignment made by the software to a specific ID in a specific frame. The user was then asked to classify the assignment as either 'correct' or 'incorrect'. If the user was unsure of the correctness of the assignment, they could skip to the next one. The process was repeated until the user had identified 500 points as either correct or incorrect. The accuracy of the tracking was measured as the ratio of correct test points to the sum of correct and incorrect test points, as determined by the human observer. This procedure samples the range of experimental conditions and behavioral states represented in each of the datasets in an unbiased manner, and provides a tracking performance estimate that can be applied and compared across experiments. Overall, anTraX performed at a level close to the human observer in all benchmark datasets (*Table 2*).

## Graph inference dramatically improves tracking performance

The main novelty of anTraX compared to other tracking solutions is the use of a tracklet graph for ID inference. This method increases the tracking performance in several ways. First, it allows identification of tracklets that are unidentifiable by the classifier, using propagation of IDs from classified tracklets. Second, it corrects classification errors by overriding low-reliability assignments made by

**Table 1.** Summary description of the benchmark datasets.
All raw videos and parameters of the respective tracking session are available for download (*Gal et al., 2020b*).

| Dataset | Species | #Animals | #Colors | #Tags | Open* ROI | Duration (hr) | Camera | FPS | Image size (pixels) | Resolution (pixels/ mm) |
|---------|---------|----------|---------|-------|-----------|---------------|--------|-----|---------------------|-------------------------|
| J16 | *Ooceraea biroi* | 16 | 4 | 2 | No | 24 | Logitech C910 | 10 | 960 × 720 | 10 |
| A36 | *Ooceraea biroi* | 36 | 6 | 2 | No | 24 | PointGrey Flea3 12MP | 10 | 3000 × 3000 | 25 |
| C12 | *Camponotus fellah* | 12 | 7 | 3 | No | 6 | Logitech C910 | 10 | 2592 × 1980 | 17 |
| C32 | *Camponotus sp.* | 28 | 6 | 3 | No | 24 | PointGrey Flea3 12MP | 10 | 2496 × 2500 | 13 |
| G6 × 16 | *Ooceraea biroi* | 6 × 16[†] | 3 | 2 | No | 1.33 | Logitech C910 | 10 | 2592 × 1980 | 17 |
| V25 | *Ooceraea biroi* | 25 | 5 | 2 | Yes | 3 | Logitech C910 | 10 | 2592 × 1980 | 17 |
| T10 | *Temnothorax nylanderi* | 10 | 5 | 4 | No | 6 | Logitech C910 | 10 | 2592 × 1980 | 17 |
| D7 | *Drosophila melanogaster* | 7 | 7 | 1 | No | 3 | PointGrey Flea3 12MP | 18 | 1056 × 1050 | 26 |
| D16 | *Drosophila melanogaster* | 16 | 4 | 2 | No | 5 | PointGrey Flea3 12MP | 18 | 1200 × 1200 | 16 |

ROI: region of interest; FPS: frames per second. *Whether or not the ants can leave the tracked region. [†]Dataset G6 × 8 is derived from six replicate colonies with eight ants each.

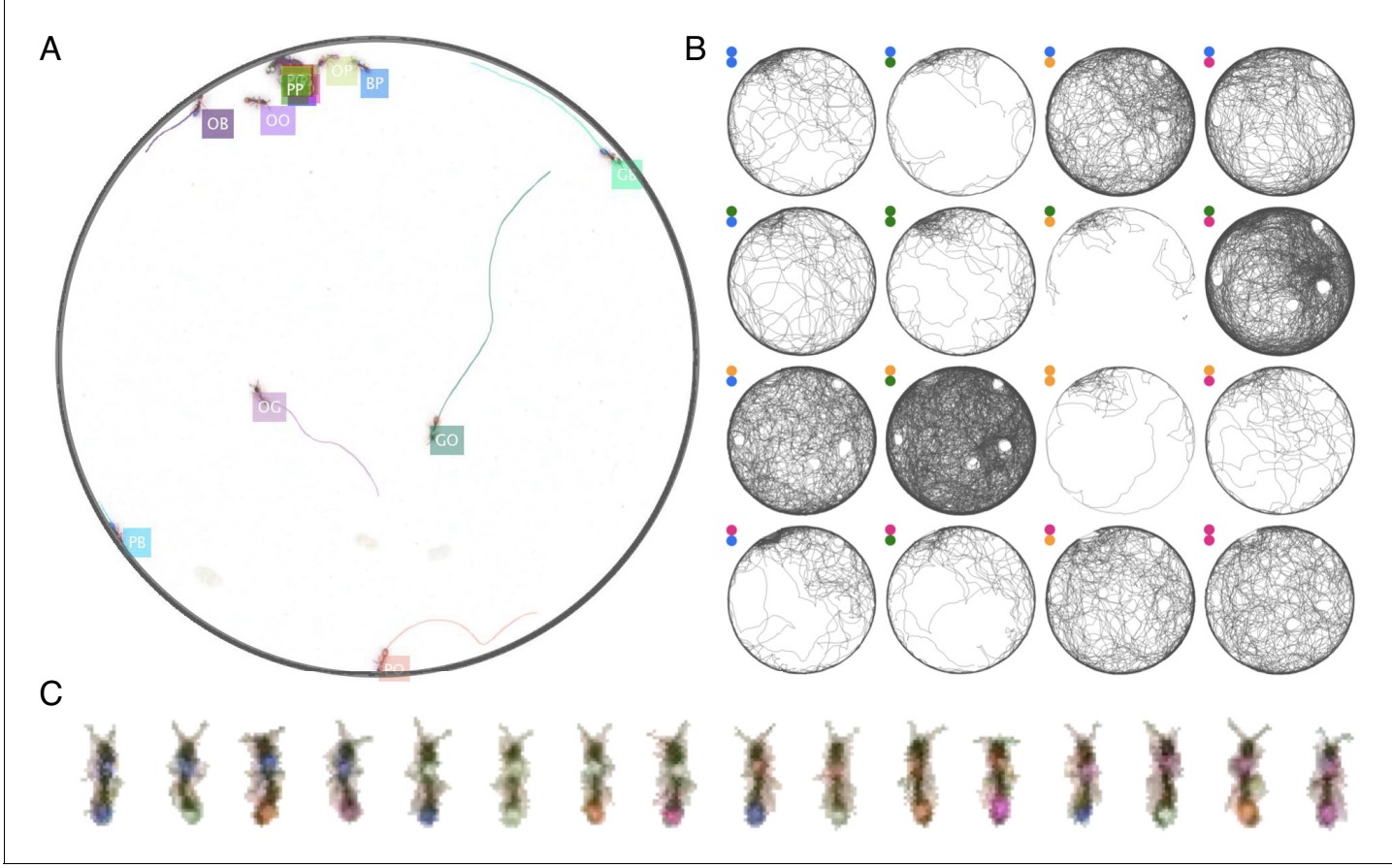

**Figure 3.** Example of anTraX tracking output, based on the J16 dataset. In this experiment, the ants are freely behaving in a closed arena that contains the nest (the densely populated area on the top left) and exploring ants. A short annotated clip from the tracked dataset is given as *Figure 3—video 1*. Tracking outputs and annotated videos of all datasets are also given in the supplementary figures and videos of this figure. (**A**) A labeled frame (background subtracted), showing the location of each ant in the colony, as well as a 'tail' of the last 10 s of trajectory. Ants that are individually segmented have precise locations. The ants clustered together have approximate locations. Labels indicate the color tag combination of the ant (e.g. 'BG' indicates a blue thorax tag and a green abdomen tag; colors are blue (B), green (G), orange (O), and pink (P)). (**B**) Individual trajectories for each ant in the colony, based on 1 hr of recording. (**C**) A cropped image of each ant from the video.

The online version of this article includes the following video and figure supplement(s) for figure 3:

**Figure supplement 1.** Tracking the V25 dataset with 25 *O. biroi* ants.
**Figure supplement 2.** Tracking the A36 dataset with 36 *O. biroi* ants.
**Figure supplement 3.** Tracking the T10 dataset with 10 *T. nylanderi* ants.
**Figure supplement 4.** Tracking the T10 dataset with 12 *C. fellah* ants.
**Figure supplement 5.** Tracking the C32 dataset with 28 *Camponotus* spec. ants, including an unmarked winged queen.
**Figure supplement 6.** Tracking the D7 dataset with seven *D. melanogaster* fruit flies.
**Figure supplement 7.** Tracking the D16 dataset with 16 *D. melanogaster* fruit flies.
**Figure supplement 8.** Tracking the G6 × 16 dataset with six colonies of 16 *O. biroi* ants each recorded and tracked in parallel.
**Figure 3—video 1.** An annotated tracking video clip from dataset J16.
https://elifesciences.org/articles/58145#fig3video1
**Figure 3—video 2.** An annotated tracking video clip from dataset V25.
https://elifesciences.org/articles/58145#fig3video2
**Figure 3—video 3.** An annotated tracking video clip from dataset A36.
https://elifesciences.org/articles/58145#fig3video3
**Figure 3—video 4.** An annotated tracking video clip from dataset T10.
https://elifesciences.org/articles/58145#fig3video4
**Figure 3—video 5.** An annotated tracking video clip from dataset C12.
https://elifesciences.org/articles/58145#fig3video5
**Figure 3—video 6.** An annotated tracking video clip from dataset C32.
https://elifesciences.org/articles/58145#fig3video6
*Figure 3 continued on next page*

*Figure 3 continued*

**Figure 3—video 7.** An annotated tracking video clip from dataset D7.

https://elifesciences.org/articles/58145#fig3video7

**Figure 3—video 8.** An annotated tracking video clip from dataset D16.

https://elifesciences.org/articles/58145#fig3video8

**Figure 3—video 9.** An annotated tracking video clip from dataset G6 × 16.

https://elifesciences.org/articles/58145#fig3video9

the classifier with IDs propagated from high-reliability tracklets. Third, it assigns IDs to multi-individual blobs and tracklets. This provides an approximate location for analysis, even when an animal cannot be individually segmented. *Table 2* and *Figure 4A–B* show the increase in assignment coverage and decrease in assignment errors following graph propagation in all benchmark datasets.

To further demonstrate the utility of graph propagation, we used data from a full, large-scale experiment. We tracked the behavior of 10 clonal raider ant colonies, each consisting of 16 ants, for 14 days. The colonies were filmed at relatively low resolution using simple webcams (Logitech C910, 960 × 720 pixels image size, 10 frames per second), similar to that of benchmark dataset J16. This dataset represents a relatively challenging classification scenario, because the tags are small, and the colors are dull. *Figure 4C–D* show a comparison of assignment rate and accuracy across the 10 replicates before and after graph propagation, with both measures improving greatly. Moreover, the assignments made by the propagation algorithm are as reliable as the assignments made directly by the classifier (*Figure 4—figure supplement 1*). The propagation algorithm is also robust to classification errors, and successfully blocks their propagation on the tracklet graph (*Figure 4—figure supplement 2*).

**Table 2.** Summary of tracking performance measures for the benchmark datasets using anTraX.

Assignment rate is defined as the proportion of all data points (the number of individuals times the number of frames) in which a blob assignment was made. In cases of closed boundary regions of interest (ROIs; in which the tracked animals cannot leave the tracked region) this measure is in the range of 0–1. In cases of open boundary ROIs (marked with asterisks; e.g., dataset V25), the upper boundary is lower, reflecting the proportion of time the ants are present in the ROI. The assignment error is an estimation of the proportion of wrong assignments (i.e. an ant ID was assigned to a blob the respective ant is not present in). As explained in the text, the estimation is done by sequentially presenting the user with a sequence of randomly sampled assignments from the dataset and measuring the proportion of assignments deemed 'incorrect' by the observer, relative to the sum of all 'correct' and 'incorrect' assignments. To calculate the error rates reported in the table, the presentation sequence continued until exactly 500 assignments were marked as 'correct' or 'incorrect', ignoring cases with the third response 'can't say'. A 95% confidence interval of the error according to the Clopper-Pearson method for binomial proportions is also reported in the table. To quantify the contribution of using graph propagation in the tracking algorithm, the analysis was repeated ignoring assignments made during the graph propagation step, and the results are reported here for comparison. A graphical summary of the performance measures is shown in *Figure 4A–B*.

| | Without graph propagation | | | With graph propagation | | |
|---|---|---|---|---|---|---|
| Dataset | Assignment rate | Assignment error | Assignment error 95% CI | Assignment rate | Assignment error | Assignment error 95% CI |
| J16 | 0.28 | 0.012 | 0.0044–0.026 | 0.93 | 0 | 0–0.0074 |
| A36 | 0.24 | 0.014 | 0.0056–0.0286 | 0.81 | 0.006 | 0.0012–0.0174 |
| C12 | 0.82 | 0 | 0–0.0074 | 0.99 | 0 | 0–0.0074 |
| C32 | 0.26 | 0.042 | 0.0262–0.0635 | 0.79 | 0.022 | 0.011–0.039 |
| G6 × 16 | 0.57 | 0.122 | 0.0946–0.154 | 0.89 | 0.078 | 0.056–0.105 |
| V25 | 0.07* | 0.058 | 0.0392–0.0822 | 0.48* | 0.012 | 0.0044–0.026 |
| T10 | 0.56 | 0.06 | 0.041–0.0845 | 0.96 | 0.018 | 0.0083–0.339 |
| D7 | 0.88 | 0 | 0–0.0074 | 0.98 | 0 | 0–0.0074 |
| D16 | 0.89 | 0.004 | 0.0005–0.0144 | 0.997 | 0 | 0–0.0074 |

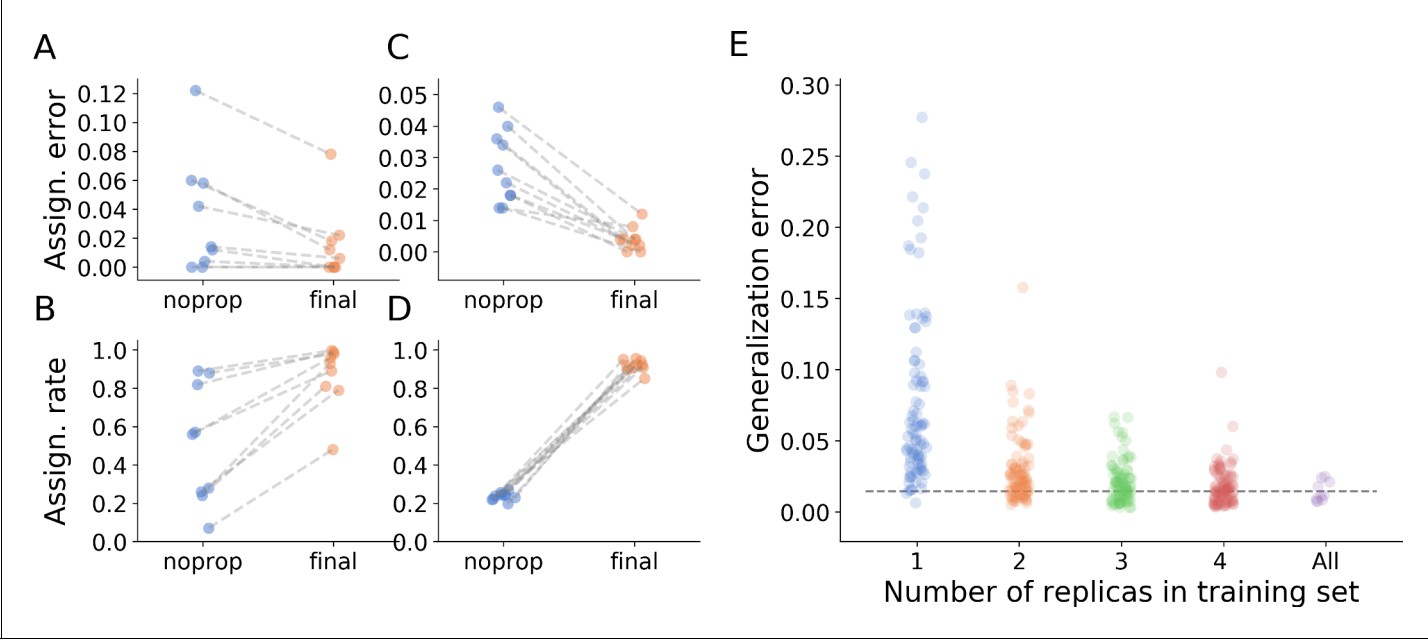

**Figure 4.** Tracking performance. (**A**) Contribution of graph inference to reduction of assignment error. The graph compares the assignment error in the benchmark datasets, defined as the rate of assigning wrong IDs to blobs across all IDs and frames in the experiment, and estimated as explained in the main text, before the graph propagation step of the algorithm (blue circles, 'noprop' category) and after the graph propagation step (orange circles, 'final' category). (**B**) Contribution of graph inference to increased assignment rate (the ratio of assignments made by anTraX to the total number of assignments possible in the experiment) in the benchmark datasets. The graph compares the assignment rate, as defined in the main text, before and after the graph propagation step (same depiction as in **A**). The performance measures for all benchmark datasets are reported in *Table 2* and *Figure 4—source data 1*. (**C–D**) Same as in **A** and **B**, calculated for a large-scale dataset described in the text (10 colonies of 16 ants, recorded over 14 days). The performance measures for all replicas are reported in *Figure 4—source data 2*. (**E**) Generalizability of the blob classifier. Each point in categories 1–4 represents the generalization error of one classifier (trained with examples from number of replicas corresponding to its category) on data from one replica that was not used for its training. The replicas were recorded under similar conditions, but using different ants, different cameras, and different experimental setups. For classifiers trained on more than one replica, the combinations of replicas were randomly chosen, while maintaining the constraint that each replica is tested against the same number of classifiers in each condition. In the category 'All', the points depict the validation error of the full classifier, trained on data from the 10 replicas. All classifiers were trained with the same network architecture, started training from a scratch model, and were trained until saturation. The dashed line represents the mean validation error for the full classifier. The list of errors for all trained classifiers are given in *Figure 4—source data 3*.

The online version of this article includes the following source data and figure supplement(s) for figure 4:

**Source data 1.** A table of performance measures, as defined in the main text, for the benchmark datasets.
**Source data 2.** A table of performance measures, as defined in the main text, for the 10-replica experiment described in the main text.
**Source data 3.** A table of generalization errors for all classifiers, as described in the caption of *Figure 4E*.
**Figure supplement 1.** Error comparison between assignments made by direct classification and assignments made by the propagation algorithm.
**Figure supplement 2.** Propagation of errors.
**Figure supplement 2—source data 1.** The count data for the histogram plotted in *Figure 4—figure supplement 2A*.
**Figure supplement 2—source data 2.** The count data for the histogram plotted in *Figure 4—figure supplement 2B*.
**Figure supplement 3.** Classification and assignment errors.

## The blob classifier generalizes well across experimental replicates

Collecting examples and training the blob classifier is the most time-consuming step in the tracking pipeline, and a good classification is essential for high-quality tracking (*Figure 4—figure supplement 3*). Ideally, a universal blob classifier would be trained to identify the same tag combination across experiments, without the need to retrain a classifier for each experiment. In reality, however, this is impractical. CNN classifiers do not generalize well outside the image distribution they were trained on, so even apparently small changes in experimental conditions (e.g. the type or level of lighting used, or the color tuning of the camera) can markedly decrease classification performance. Nevertheless, when experiments are conducted using similar conditions (e.g. study organism, marking technique, experimental setup, etc), it is possible to construct a classifier that will generalize

across these experiments with minimal or no retraining. This enables construction of efficient tracking pipelines for high-throughput and replicate experiments, without the need for additional manual annotations.

We assessed the generalizability of blob classifiers with the 10 replicates of the experiment described in the previous section. We trained a classifier on examples from one replicate, and then used it to classify blobs sampled from the other replicates. We similarly evaluated the performance of classifiers trained with examples from two, three, and four replicates, and compared the results to the performance of a classifier trained on examples from all replicates. The comparison shows that, despite variability in animal shape and behavior, tagging process, cameras, and experimental setups across replicates, the classifier performs remarkably well (*Figure 4E*). Moreover, when a classifier is trained with an example set obtained from as few as two replicates, it performs similarly well as a classifier trained with examples from all replicates. Obviously, the generalizability of this result will depend on how well conditions are standardized between replicates or experiments. Nevertheless, it demonstrates that robust behavioral tracking pipelines can be constructed with minimal retraining.

## anTraX can be combined with JAABA for efficient behavioral annotation of large datasets

While direct analysis of the tracking output is a possibility, phenotyping high-throughput experiments and extracting useful information from large-scale trajectory data beyond very simple measures are challenging and impractical. In recent years, the field of computational ethology has shifted to the use of machine learning, both supervised and unsupervised, for analyzing behavioral data (*Todd et al., 2017*; *Egnor and Branson, 2016*; *Datta et al., 2019*). One of the most useful and mature tools is JAABA, a package for behavioral annotation of large datasets using supervised learning (*Kabra et al., 2013*; *Robie et al., 2017b*). In short, JAABA projects trajectory data onto a high dimensional space of per-frame features. The user then provides the software with a set of examples for a certain behavior, and the software trains a classifier to find all occurrences of that behavior in a new dataset. anTraX includes functions to generate the per-frame data in a JAABA-compatible way. In addition to the original list of JAABA features, a set of anTraX-specific features is also generated (see online documentation for details). Beyond useful information about the appearance and kinematics of the tracked animals, these extra features provide information about whether an animal was segmented individually or was part of a multi-animal blob. This enables JAABA to learn behaviors that can only be assigned to individually segmented animals, such as those that depend on the velocity of the animal. The user can then label examples and train a classifier in the JAABA interface. This classifier can then be used to analyze entire experiments using the anTraX interface.

To demonstrate the power of this approach, we present two examples of using JAABA together with anTraX. In the first example, we train a classifier to detect *O. biroi* ants carrying a larva while walking. *O. biroi* ants carry their larva under their body, in a way not always obvious even to a human observer (*Figure 5A*, *Figure 5—video 1*). By using subtle changes in the ants' appearance and kinematics, JAABA is able to classify this behavior with >93% accuracy (tested on a set of annotated examples not used for training). An example of trajectories from a 30 min period annotated with JAABA is shown in *Figure 5B*.

In the second example, we used JAABA to classify the behavior of ants during periods when they are not moving. We trained a classifier to detect four distinct behaviors (*Figure 5—video 2*): *rest*, in which the ant is completely immobile; *local search*, in which the ant does not move but uses its antennae to explore the immediate environment; *self-grooming*, in which the ant stops to groom itself; and *object-interaction*, in which the ant interacts with a non-ant object such as a piece of food, a larva or a trash item. JAABA was able to identify these behaviors with >92% accuracy. *Figure 5C* shows the spatial distribution of the classified behaviors during all periods where an ant stops walking for more than 2 s in a 60-min experiment, across all ants in the colony.

## anTraX can be combined with DeepLabCut to augment positional data with pose tracking

Much attention has recently been given to tracking the relative position of animal body parts, taking advantage of the fast progress in machine learning and computer vision (*Pereira et al., 2019*; *Mathis et al., 2018*; *Graving et al., 2019*). This allows for the measurement and analysis of aspects

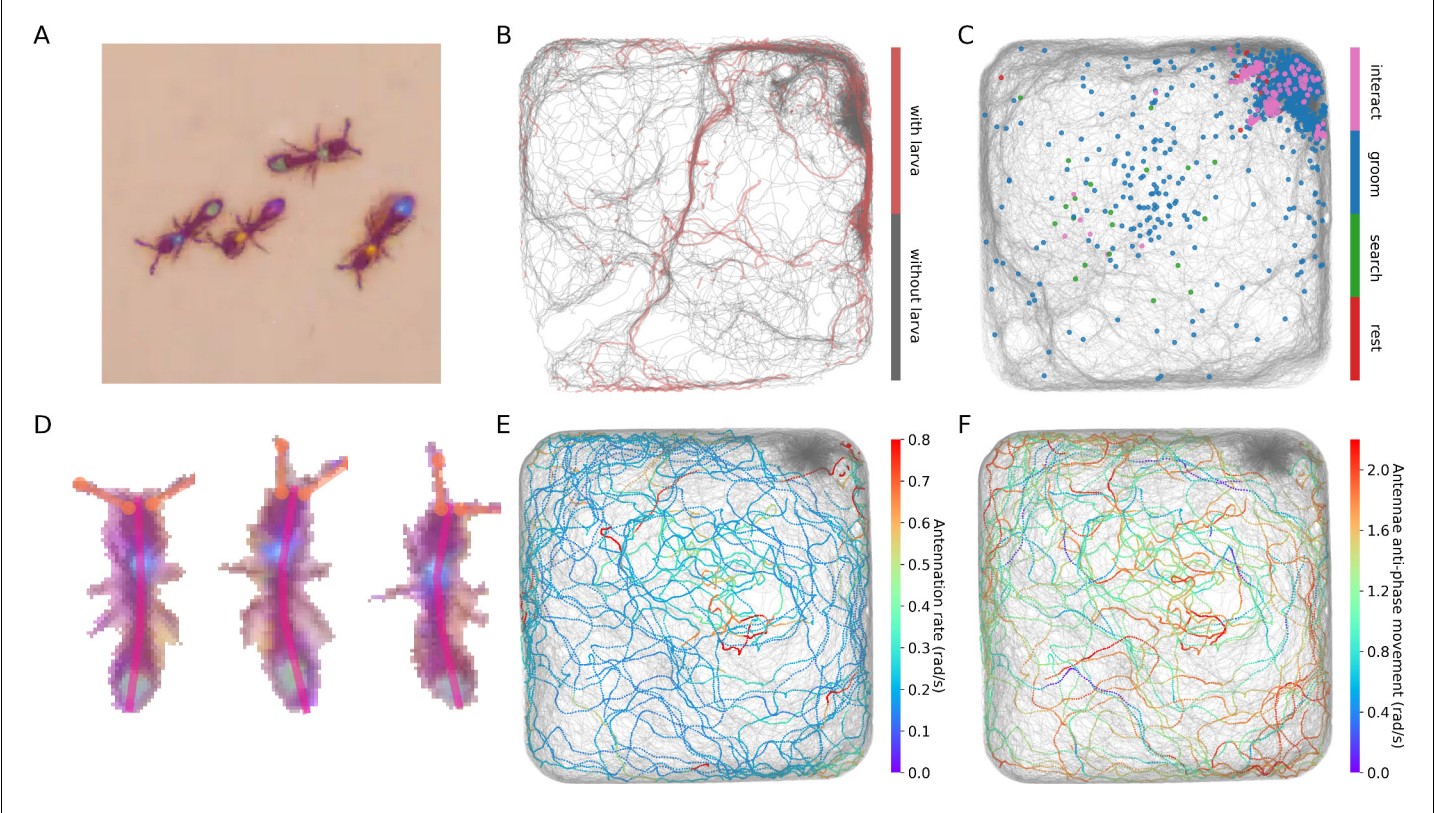

**Figure 5.** Interfacing anTraX with third party behavioral analysis packages for augmenting tracking data. (A) Ants carrying a larva while they move (green/green and yellow/blue) can be difficult to distinguish from ants not carrying larvae (blue/green and yellow/purple), even for a human observer. *Figure 5—video 1* shows examples for ants walking with and without a larva. (B) However, using labeled examples to train a classifier, JAABA can reliably distinguish ants walking while carrying a larva from ants walking without one from anTraX tracking output. Shown here is a 30 min segment from the A36 dataset, where trajectories classified by JAABA as ants carrying a larva are plotted in red on the background of all other tracks (in gray). (C) Classifying stops using JAABA. The plot shows a 60 min segment from the A36 experiment, where all stops longer than 2 s are marked with a colored dot. The stops are classified into four categories: rest (red), local search (green), self-grooming (blue), and object-interaction (e.g. with a food item; pink). *Figure 5—video 2* shows examples of stops from all types. (D) Applying a simple DeepLabCut model to track the ants' antennae and main body axes, shown on segmented ant images from dataset A36. *Figure 5—video 3* shows an animated tracking of all ants in the colony. (E–F) Using the results from DeepLabCut to track the behavior of an ant along its trajectory. A one-hour trajectory of one ant from dataset A36 is shown on the background of the tracks of all other ants in the colony in that period (in gray). In E, the focal trajectory is colored according to the total rate of antennal movement (measured in angular velocity units rad/s). In F, the focal trajectory is colored according to how much the antennae move in-phase or anti-phase (measured in angular velocity units rad/s). Together, these panels show the behavioral variability in antennal movement patterns.

The online version of this article includes the following video(s) for figure 5:

**Figure 5—video 1.** Examples of short video clips from dataset A36 in which some ants walk carrying a larva, while others walk without a larva.
https://elifesciences.org/articles/58145#fig5video1

**Figure 5—video 2.** Examples of short video clips from dataset A36 showing the four types of stop behavior.
https://elifesciences.org/articles/58145#fig5video2

**Figure 5—video 3.** Pose-tracking of all ants in dataset A36 using anTraX in combination with DeepLabCut.
https://elifesciences.org/articles/58145#fig5video3

of an animal's behavior beyond what is extractable from its trajectory. Although these tools can in principle be directly applied to videos with multiple individuals (*Iqbal et al., 2017*; *Insafutdinov et al., 2016*), they are still not mature enough for large-scale use. A more reasonable approach it to combine individual animal pose tracking with a track-and-crop step (see discussion within *Graving et al., 2019*). To track body parts of individual animals within a group or a colony, we took advantage of the fact that anTraX segments and crops the images of individual animals as part of its workflow, and included an option to run pre-trained DeepLabCut models (*Mathis et al., 2018*) on these images, without the need to export the data in a DeepLabCut-readable format

(which would have resulted in a heavy computational overhead). This way, the position of the tracked body parts relative to the animal's centroid are returned together with the spatial location of the centroid. For training such a model, anTraX enables exporting cropped single animal videos that are loadable into the DeepLabCut user interface. Currently, this is only supported for single-animal tracklets, where animals are segmented individually.

Of course, the ability to perform accurate and useful pose estimation depends on the resolution at which animals appear in the video. To demonstrate the potential of this approach, we trained a simple DeepLabCut model to track the main body axis and antennae positions of ants from benchmark dataset A36. *Figure 5D* and *Figure 5—video 3* show examples from the segmented and cropped images of the individual ants in the videos.

Ants use different antennation patterns to explore their environment (*Draft et al., 2018*), and the ability to track these patterns in parallel to their movement in space can contribute to our understanding of their sensory processing during free behavior. We used the pose tracking results to visualize the different modes of antennae movement used by the ants to explore their environment. Figure panels *Figure 5E and F* show the total movement rate and the relative phase of the two antennae along the trajectory of one ant in a 1-hr segment of the experiment, respectively, demonstrating the variability and richness inherent to these patterns.

## Discussion

anTraX is a new algorithm and software package that provides a solution for a range of behavioral tracking problems not well addressed by available methods. First, by using a deep neural network for image classification, it enables the tracking of insects that are individually marked with color tags. While color tags have been used successfully for behavioral analysis for decades in a wide range of social insects, and in many species they are the only practical type of marker, their use has been severely limited by the lack of automation. Second, unlike other existing approaches, it handles cases where insects tightly aggregate and are not segmentable, as well as cases where the tags are obscured. This is achieved by representing the tracking data as a directed graph, and using graph walks and logical operations to propagate information from identified to unidentified nodes. Third, anTraX handles very long experiments with many replicate colonies and minimal human oversight, and natively supports parallelization on computational clusters for particularly large datasets. Finally, anTraX can easily be integrated into the expanding ecosystem of open-source software packages for behavioral analysis, making a broad range of cutting-edge ethological tools available to the social insect community. anTraX is an open-source software and conveniently modular, with each step of the algorithm (segmentation, linking, classification, and propagation) implemented as a separate module that can be easily augmented or replaced to fit experimental designs that are not well handled by the current version of the algorithm. For example, the traditional background subtracted segmentation can be replaced with a deep learning-based semantic segmentation, that is training and using a classifier to distinguish pixels of the image as belonging to either background or foreground (*Rajchl et al., 2017*; *Moen et al., 2019*; *Badrinarayanan et al., 2017*). This can potentially allow analysis of field experiments with natural backgrounds, or experiments with non-static backgrounds, such as videos taken with a moving camera. Another possible extension is an informed 'second pass segmentation' step, where multi-animal blobs are further segmented into single-animal blobs, taking into account the composition of the blob (number and IDs of animals). Knowing the composition of the blob provides a method to algorithmically validate the segmentation, allowing a 'riskier' segmentation approach. Another approach to locate animals in aggregations more precisely is to use neural network-based detection of the tags themselves. This method has successfully been used for bees tagged with fiducial markers inside a hive (*Wild et al., 2018*). Having a record of the composition of tracklets and blobs also paves the way for performing image-based behavioral analysis of interactions (*Dankert et al., 2009*; *Klibaite et al., 2017*; *Klibaite and Shaevitz, 2019*), or constructing specialized image classifiers for interaction types (e.g. allogrooming, trophallaxis, aggression, etc). Lastly, a newer generation of pose-estimation tools, including SLEAP (*Pereira et al., 2020*) and the recent release of DeepLabCut with multi-animal support, enable the tracking of body parts for multiple interacting animals in an image. These tools can be combined with anTraX in the future to extend pose tracking to multi-animal tracklets, and to augment positional information for individual animals within aggregations.

In summary, anTraX fills an important gap in the range of available tools for tracking social insects, and considerably expands the range of trackable species and experimental conditions. It also interfaces with established ethological analysis software, thereby making these tools broadly accessible for the study of social insects. anTraX therefore has the potential to greatly accelerate our understanding of the mechanisms and principles underlying complex social and collective behavior.

## Acknowledgements

We thank Z Frentz for invaluable advice during the development of the algorithm. T Kay, O Snir, S Valdés Rodríguez and L Olivos-Cisneros helped in collecting and marking ants and flies. Y Ulrich, G Alciatore, and V Chandra tested the software and shared datasets for benchmarking. Research reported in this publication was supported by the National Institute of General Medical Sciences of the National Institutes of Health under Award Number R35GM127007 to DJCK. The content is solely the responsibility of the authors and does not necessarily represent the official views of the National Institutes of Health. This work was also supported by a Searle Scholar Award, a Klingenstein-Simons Fellowship Award in the Neurosciences, a Pew Biomedical Scholar Award, and a Faculty Scholars Award from the Howard Hughes Medical Institute to DJCK. AG was supported by the Human Frontiers Science Program (LT001049/2015). JS was supported by a Kravis Fellowship awarded by Rockefeller University. This is Clonal Raider Ant Project paper #14.

## Additional information

### Funding

| Funder | Grant reference number | Author |
| --- | --- | --- |
| National Institute of General Medical Sciences | R35GM127007 | Daniel JC Kronauer |
| Searle Scholars Program | | Daniel JC Kronauer |
| Klingenstein-Simons | Fellowship Award in the Neurosciences | Daniel JC Kronauer |
| Pew Charitable Trusts | | Daniel JC Kronauer |
| Howard Hughes Medical Institute | | Daniel JC Kronauer |
| Human Frontier Science Program | LT001049/2015 | Asaf Gal |
| Rockefeller University | Kravis Fellowship | Jonathan Saragosti |

The funders had no role in study design, data collection and interpretation, or the decision to submit the work for publication.

### Author contributions

Asaf Gal, Conceptualization, Data curation, Software, Formal analysis, Validation, Investigation, Visualization, Methodology, Writing - original draft, Project administration, Writing - review and editing; Jonathan Saragosti, Conceptualization, Data curation, Software, Investigation, Methodology, Writing - review and editing; Daniel JC Kronauer, Conceptualization, Resources, Supervision, Funding acquisition, Project administration, Writing - review and editing

### Author ORCIDs

Asaf Gal https://orcid.org/0000-0003-0834-2649
Daniel JC Kronauer https://orcid.org/0000-0002-4103-7729

### Decision letter and Author response

Decision letter https://doi.org/10.7554/eLife.58145.sa1
Author response https://doi.org/10.7554/eLife.58145.sa2

## Additional files

### Supplementary files

• Transparent reporting form

### Data availability

Benchmark datasets, together with the anTraX configuration files used for the analyses, have been deposited in Zenodo under record number 3740547. anTraX software version 1.02 has been deposited in Zenodo under record number 3774487. Source code and binaries for anTraX are also available at https://github.com/Social-Evolution-and-Behavior/anTraX (copy archived at https://archive.softwareheritage.org/swh:1:rev:98813386637592f55dbaf426803a9ff0f60ee711/). Online documentation and a detailed user manual are available at https://antrax.readthedocs.io. Source data files have been provided for the performance analysis in Figure 4.

The following datasets were generated:

| Author(s) | Year | Dataset title | Dataset URL | Database and Identifier |
|---|---|---|---|---|
| Gal A, Saragosti J, Kronauer DJC | 2020 | anTraX: high throughput video tracking of color-tagged insects (software) | https://doi.org/10.5281/zenodo.3774487 | Zenodo, 10.5281/zenodo.3774487 |
| Gal A, Saragosti J, Kronauer DJC | 2020 | anTraX: high throughput video tracking of color-tagged insects (benchmark datasets) | https://doi.org/10.5281/zenodo.3740547 | Zenodo, 10.5281/zenodo.3740547 |

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

## Appendix 1

### Detailed description of the anTraX algorithm

The anTraX algorithm consists of three main steps (*Appendix 1—figure 1*). In the first step, we detect the tracked animals in each frame of the video and organize the extracted *blobs* into trajectory pieces we term *tracklets*. As we will detail below, these tracklets are in turn linked together to form an acyclic directed graph we name the *tracklet graph*. The second step of the algorithm is tracklet classification, in which identifiable tracklets are classified based on the color tag information present in their blobs. In the third step of the algorithm, we use the topology of the tracklet graph to propagate identity information from the classified tracklets to the entire set of tracklets.

In this appendix, we detail each part of the algorithm and fully describe its various computational steps and parameters. A practical tutorial for running the software and using its graphical interface can be found in the online documentation.

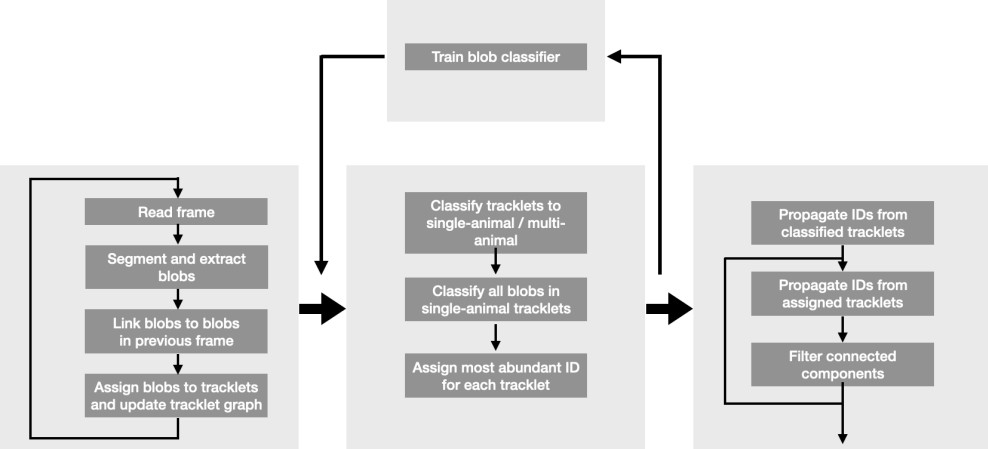

**Appendix 1—figure 1.** Flow diagram of the anTraX algorithm.

## 1. Creating the tracklet graph

### 1.1 Creating a background image

anTraX uses background subtraction for segmentation. Although using a static background is somewhat limiting in designing and performing experiments (requiring a static environment and a static camera), and it is possible to segment images for tracking without this step if there is a decent contrast between the objects and the background, background subtraction has the advantage of giving a stable object segmentation that simplifies later steps.

For creating a background image, anTraX uses random sampling of frames from the entire duration of the experiment, or from a segment defined by the user (*Appendix 1—figure 2A,B*). The number of frames $n_B$ is configurable, and the background $I^{BG}$ is computed by applying either a per-pixel median or max operator:

$$I^{BG}(i,j,c) = med\{I^{t_b}(i,j,c)\}_{b=1}^{n_B} \tag{1A}$$

$$I^{BG}(i,j,c) = max\{I^{t_b}(i,j,c)\}_{b=1}^{n_B} \tag{1B}$$

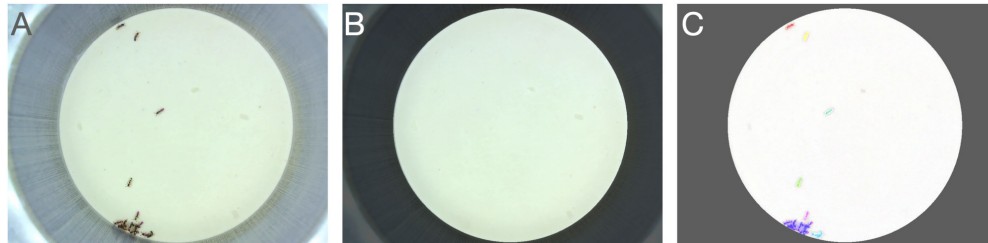

**Appendix 1—figure 2.** Background creation. (**A**) An example raw frame. (**B**) A background frame generated using a median function. Regions outside the ROI mask are dimmed. (**C**) Full segmented frame.

Where $t_b$ is a randomly drawn timepoint in the experiment, $I^{t_b}$ is the corresponding frame, $i$ and $j$ are the image coordinates, and $c$ is the color channel index.

Generally, the median operation is useful in cases where animals are active enough to have each pixel in the image free of animals for at least half the frames. Otherwise, the max operation gives better results. The anTraX GUI enables the user to test and optimize the parameters in the background image creation step.

## 1.2 Creating an ROI mask

Typically, tracking should be performed only in part of the image, either because the animals to be tracked are confined to a region smaller than the image, or because the user cares about behavior in a small region of interest (ROI). The ROI mask $I^{ROI}$ (*Appendix 1—figure 2B*) is a binary image with the same dimensions as the video frames, which is 1 in regions to be tracked and 0 in regions to be ignored.

The anTraX GUI includes a utility to create the mask by drawing shapes to be included or excluded on a frame.

## 1.3 Image segmentation

The first step in analyzing each frame is segmenting it into *blobs* (*Appendix 1—figure 2C*, *Appendix 1—figure 3*): contiguous regions of the frame that significantly differ from the background and correspond to individual animals or tightly clustered groups of animals. Segmentation is done by first subtracting the image from the background (using the fact that the animals are dark and tracked on a light background), then converting the difference to a grayscale image (*Appendix 1—figure 3A-B*), and comparing to a user defined threshold $\theta_s$ and the ROI mask to produce a binary image:

$$I_t^1(i,j,c) = I^{BG}(i,j,c) - I_t(i,j,c) \tag{2}$$

$$I_t^2(i,j) = \frac{1}{3}\sum_c I_t^1(i,j,c) \tag{3}$$

$$I_t^{bw}(i,j) = \begin{cases} 1, & I_t^2(i,j) \cdot I^{ROI}(i,j) > \theta_s \\ 0, & else \end{cases} \tag{4}$$

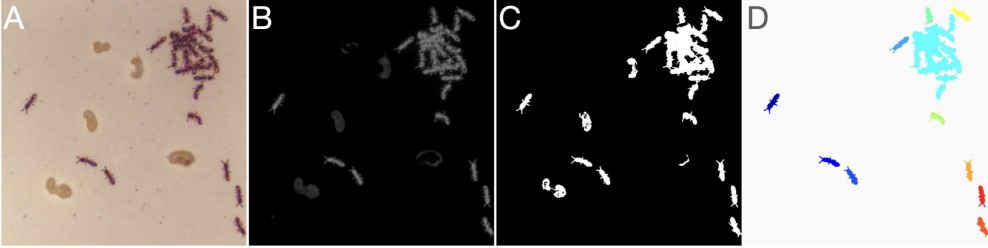

**Appendix 1—figure 3.** Image segmentation. (**A**) Raw image. (**B**) Background subtracted grayscale

image. (**C**) Unfiltered binary image. (**D**) Final segmented image after morphological operations and blob filtering. Each separate blob is shown in a different color.

The resulting binary image (*Appendix 1—figure 3C*) will then undergo optional morphological operations (image closing, image opening, hole-filling, convex hull filling) that, depending on the specific conditions of the experiment, are useful for noise reduction.

Blobs (connected components; using the eight-connectivity criterion) are then extracted from the final binary image. For each detected blob, we register the coordinates of its centroid, its area, its maximal intensity (in the $I_t^2$ grayscale image) and the parameters of the best fitted ellipse (orientation, eccentricity, and major axis length). Blobs are then optionally filtered by minimal area and minimal intensity criteria (*Appendix 1—figure 3D*).

The anTraX GUI allows the user to test and configure all the segmentation parameters.

### 1.4 Linking blobs across frames

After blobs are extracted from a frame, the next step in the algorithm is to link them to the blobs in the previous frame (*Appendix 1—figure 4A-E*): a link between a blob in frame $t$ and a blob in frame $t - 1$ implies that some or all of the individual animals that belong to the first blob, are present in the second one. A blob can be linked to one blob (the simplest case, where the two blobs have the same composition), to a few blobs (where animals leave or join the blob), or to none (suggesting the animals in the blob were not detected in the other frame). Relying on the fact that videos were recorded at a frame rate high enough that blobs corresponding to the same individuals will overlap in consecutive frames even when the tracked animals are moving at their maximum possible speed (for *O. biroi* ants, for example, 10 frames per second is sufficient), the most accurate method to link blobs is *Optical Flow*, which takes into account the actual pixel content of the image. It is, however, a computationally expensive algorithm, and running it on full frames is not practical for long, high-resolution videos. On the other hand, simpler and commonly used methods, such as the popular Munkres linear assignment algorithm (the Hungarian algorithm, *Munkres, 1957*) are prone to errors in dense problems such as those we aim to solve, and often require considerable amount of manual correction after automated tracking.

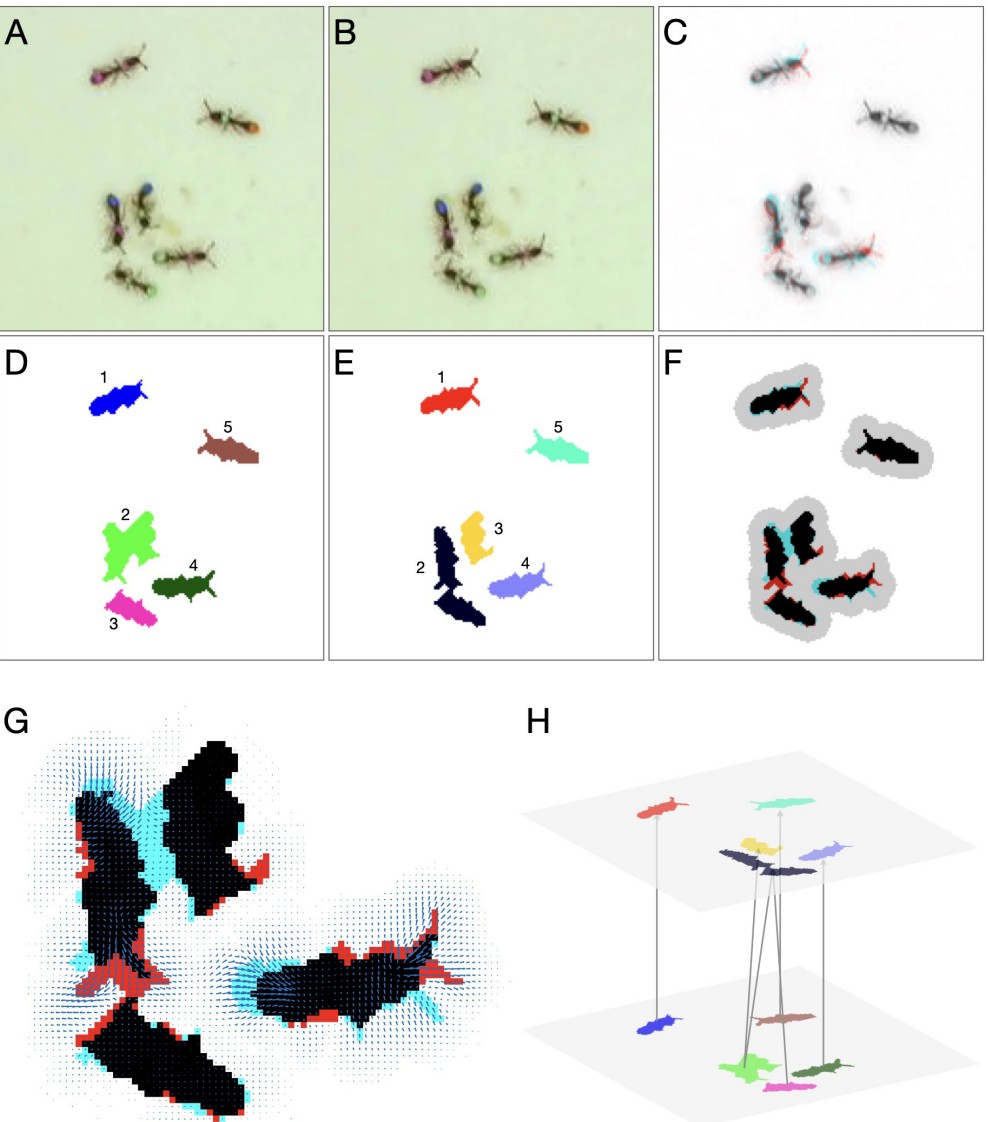

**Appendix 1—figure 4.** Detailed linking example. (**A–B**) Raw images of the first and second frame, respectively. (**C**) Color blend of the frames, showing the displacement of the ants between frames. (**D–E**) Segmentation of the first and second frame, respectively. (**F**) Segmentation blend. Also shown is the clustering of the blobs into linking problems (gray background). The two upper problems are trivial, and no assignment algorithm is required. The problem at the bottom will be solved using optical flow. (**G**) Optical flow for the bottom problem in **F**. Arrows represent the estimated translation of the pixels. (**H**) Final linking between the blobs based on optical flow.

In sophisticated tracking solutions, the distance-based cost function that underlies the linear assignment is corrected with predictive modeling of the animals' behavior, or with other distinguishing features of the animals such as shape, orientation, and appearance. These, however, are often problem-specific and do not generalize well across tracking problems. We chose to implement a dynamic approach, in which the linking method is chosen based on the difficulty of assignment. The linking step begins with dividing the linking problem into a few independent subproblems, by using a maximal linking distance ($d_{link}$), which by default is set to twice the maximal velocity $v_{max}$ times the inter frame time interval. Practically, this is done by creating a binary image, defined as the pixel-wise logical OR of the two segmented binary frames, and dilating it using a disk with a radius that equals to $d_{link}$ (*Appendix 1—figure 4f*). The resulting image is then divided into connected components, and all the blobs that overlap with each component are treated as an independent

subproblem. For each subproblem we choose the appropriate linking method: (i) a problem with one or more blobs in one of the frames and no blobs in the other results in no links, (ii) a problem with exactly one blob in each of the frames will link the blobs with no further processing, (iii) otherwise, a small region containing only the blobs in the subproblem will be cropped from each of the frames, and an optical flow assignment will be performed (*Appendix 1—figure 4G*).

For solving a subproblem using optical flow, we do the following: We first crop a region from the two frames, corresponding to the bounding box of the subproblem's connected component. This region includes all of the blobs that belong to this subproblem, but no others. We then compute the optical flow field between the two cropped frames using the Horn-Schunck method (*Horn and Schunck, 1981*). Next, we define the *Flow Number*, $n^{of}(a, b)$, for each pair of blobs across the two frames as the number of flow field vectors pointing from blob $a$ in frame $t - 1$ to blob $b$ in frame $t$. The flow number is an estimate of the number of pixels in the blob $a$ that have moved to blob $b$ in the consecutive frame. For each pair, if the flow number is greater than a threshold number $\theta_{of}$, the blobs are linked (*Figure 4H*). The threshold number defaults to a third of the minimal size of a single animal in pixels and can be configured using the anTraX graphical interface.

Once again, all the parameters of the linking step can be configured and tested in the anTraX GUI.

## 1.5 Updating the tracklet graph

As defined above, a blob can correspond to an arbitrary number of tracked individuals. Instead of trying to break these blobs down into individual animals, our tracking approach relies on registering the transition of individuals between blobs that possibly contain multiple animals. For this purpose, we define the *tracklet* as a list of linked blobs in consecutive frames that have the same composition of individuals. In other words, no animal has left or entered the group between the first and last frame of the tracklet (*Figure 1* in the main text).

After linking the blobs in frame $t$ to those in frame $t - 1$, the tracklets are updated in the following way:

1. A blob in the current frame $t$ that is not linked to any blob in the previous frame $t - 1$ will 'open' a new tracklet.
2. A blob in the previous frame that is not linked to any blob in the current frame will 'close' its tracklet.
3. If two blobs in the previous and current frames are exclusively linked, the current blob will be added to the tracklet of the previous blob.
4. Whenever blobs in the current or previous frame are connected to more than one blob, the tracklets of the linked blobs in the previous frames will 'close', and new tracklets will 'open' for the blobs in the current frame. In these cases, the linking between the blobs across different tracklets will be registered as a link between the tracklets. In cases where a tracklet has its last blob linked to the first blob of a different tracklet, the former is defined as the *parent* tracklet, and the latter as its *child* tracklet.

Although the linking and tracklet construction processes are very conservative, errors can still occur when the assumptions of the algorithm are violated. For example, in benchmark dataset J16, in which the behavior of a 16 ant colony is recorded in an uncovered arena surrounded by Fluon-coated walls, ants sometimes climb on the arena's walls and fall down on top of another ant, hence violating the maximal linking distance assumption. In such a case, the tracklet corresponding to the climbing ant will end without parenting a child tracklet, while the tracklet of the second ant will contain one ant in its first part and two ants in its second part. In the analyzed dataset, such linking errors occur very rarely (less than 0.05% of the tracklets), and in most cases will not lead to classification errors, due to the robustness of the ID propagation step to such errors (section 3).

Upon closing of a tracklet, the blob orientation has a $\pm\pi$ ambiguity as a result of the definition of the orientation as that of the best fit ellipse, which is not consistent along the tracklet (for each blob, the orientation is set independently of the other blobs in the tracklet by MATLAB's blob analysis algorithm). We use a method adapted from *Branson et al., 2009* to disambiguate the orientation. In short, this method uses the fact that whenever the tracked animal is moving fast, we can reliably assign the correct orientation in the moving direction and propagate this assignment to the entire tracklet by using dynamic programing. In tracklets where the animal is not moving fast enough, the

result can be incorrect, but it is at least consistent along the tracklet. Most of these cases will be corrected later after the tag identification step. Multi-animal tracklets generally do not have a meaningful orientation.

The end result of this part of the algorithm, after processing all frames in the video, is an acyclic directed graph containing references to all tracklets and blobs in the experiment.

## 2. Classifying tracklets

### 2.1 Color correction

The actual RGB values of the color tags are highly sensitive to changes in illumination and variability in camera sensors, both between experiments, and within an experiment as a function of time and location. These sources of variability can adversely affect the performance of the tracklet classifier. To overcome this problem, at least partially, we include the option of applying a *color correction* step on images before classification (*Appendix 1—figure 5*). To do so, we use a white reference frame $W$, which is an image of a white or gray surface taken using the same conditions as the videos. The color corrected frame is then:

$$I^W(i,j,c) = \frac{I(i,j,c)}{W(i,j,c)} \tag{5}$$

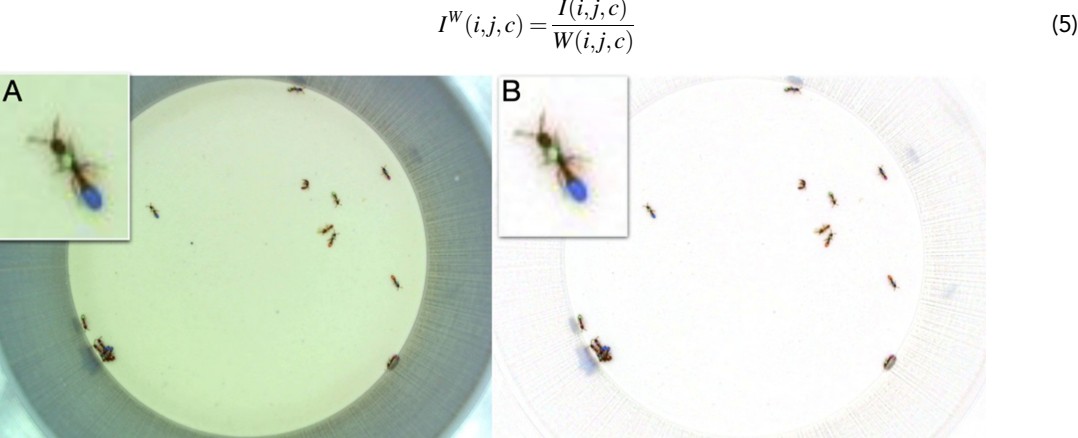

**Appendix 1—figure 5.** Color correction. (**A**) The original frame. (**B**) The color corrected frame. Insets show a zoomed in view of a focal ant. The color correction removes the green bias in the original frame and enhances the color segmentation.

Pixel values that exceed the pixel value range are truncated.

In cases where the tracking background approximates a homogenous white surface, as is the case with all the benchmark datasets, the white reference can be automatically generated by anTraX by filtering the background image with a 2D Weiner filter. In other cases, a white reference image can be taken in the experimental setup before or after the experiment.

### 2.2 Training a blob classifier

Classifying a tracklet begins with classifying the individual blobs it contains. To do so, we train a convolutional neural network (CNN) image classifier using TensorFlow (*Abadi et al., 2016*). To create a classifier, the user has to supply a list of possible labels. Typically, this will be the list of IDs (unique tag combinations) in the experiment, plus optional labels for non-animal objects that can be detected in the videos (e.g. larvae, food items, etc). One of the limitations of using CNNs for classification is the high rate of false positives, that is, blobs that are assigned an ID even though they are not identifiable. To overcome this, we add a special label for unrecognizable blobs, which are treated as a separate class (labeled as 'Unknown' or 'UK').

To train the classifier, we collect a set of example images for each classifier label (*Appendix 1— figure 6*). This can be done easily using an anTraX GUI app (see Online Documentation for details).

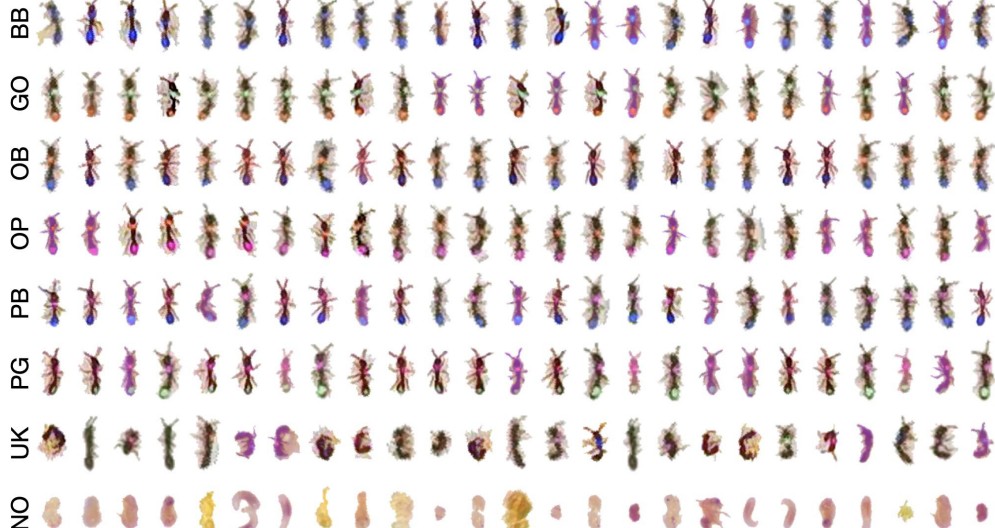

**Appendix 1—figure 6.** An example subset from a training set. Shown are examples from six ant IDs with a total of four tag colors. The UK label represents ant images that are not classifiable to the human eye. The NO label represents segmented objects that are not ants (food items, larvae, etc). To allow the classifier to generalize well, it is important that the variability of the training set captures the variability in the experiment, and includes images of ants in various poses, lighting conditions, and across experimental replicates.

In short, the GUI presents the user with all the blob images from a random tracklet. The user can then select the appropriate ID and choose to either export all images into the training set, or to select only a subset of images (useful if not all blobs in the tracklet are recognizable). In many cases, especially in social insects, where behavioral skew can be considerable, some animals are rarely observed outside an aggregation. It is therefore challenging to collect examples for them using a random sampling approach. One solution to this problem, which is the recommended one for high throughput experiments, is to pool data from several experiments into one classifier as discussed in the main text. Another solution, in case this is not possible, is to scan the video for instances in which the focal animal leaves the group, and 'ask' the GUI for tracklets from this segment of the video. Alternatively, one can run a first pass of training and classification using the available examples, and then ask the GUI to display only unclassified tracklets, increasing the probability of spotting the missing animal. The resulting example set augmented using various transformations (flipping, rotations, shearing, and brightness and color shifts; *Appendix 1—figure 7*). Some of these transformations are only applicable in certain cases (e.g. horizontal flipping will only be valid for cases where tags have a horizontal symmetry), and some are range-configurable through the anTraX interface. It is important to tune these range parameters appropriately, because there is no point in training the classifier on images that cannot actually occur in the real data. This will only slow down training and reduce performance. For example, rotations are applied by default in the range of ±15°, as we found that this value captures the variability in head orientation relative to blob orientation in most of our datasets well. For animals with low eccentricity, higher values for this parameter will produce better generalization.

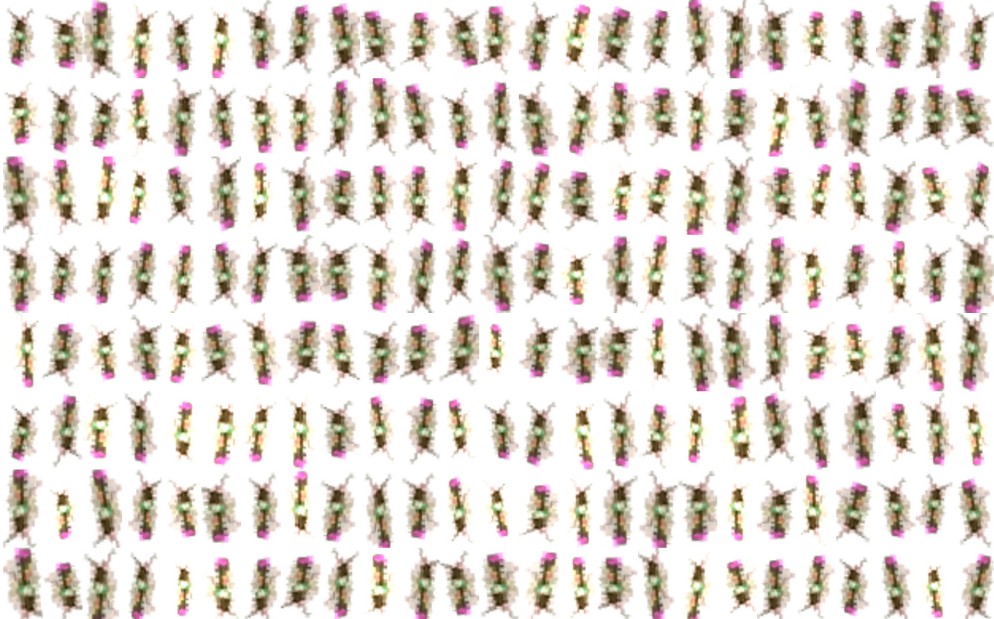

**Appendix 1—figure 7.** Dataset augmentation using TensorFlow's intrinsic mechanism for image transformation on a single example image to generate a larger training dataset.

As usual with supervised classifiers, there is a tradeoff between the complexity of the classifier (the size and architecture of the network), and its performance, training time, and the optimal size of the training dataset. anTraX contains a few CNN architectures that we have found to work well with our data. However, it can also use an arbitrary, user-defined architecture (see Online Documentation for details).

## 2.3 Filtering tracklets for classification

Once the blob classifier is trained, it can be applied to the tracklets of the experiment. Because direct classification is only meaningful for tracklets that represent individual animals, we first filter the tracklet list to identify possible single-animal tracklets. To do so, we use the typical size range of individual animals (interactively adjustable in the anTraX GUI). A tracklet whose average blob area falls within that range is considered a possible single-animal tracklet, and is passed on to the classifier. Although this filtering method is not perfect, it rarely leads to false negatives (single-animal tracklets with average blob size outside of the specified range). If the rate of false positives is high (which is usually the case in problems with high size variability between individuals), it is useful to include a separate class for multi-animal blobs.

For performance reasons, this filtering is done during the blob tracking step, and the images constituting possible single-animal tracklets are saved separately to disk, thus avoiding the need to extract them again from the videos. It is therefore important to set the single animal size range before running the tracking.

## 2.4 Classifying single-animal tracklets

To classify a possible single-animal tracklet, we perform the following steps:

1. The blob classifier is applied to each blob in the tracklet. The output of the classifier for each tracklet is a matrix of likelihoods, $\mathfrak{L}_{kl}$, that is, the probability of blob $k$ belonging to class $l$ given its image. We define the most probable label for a blob as:

$$l_k^* = argmax_l(\mathfrak{L}_{kl}) \tag{6}$$

2. If the most likely label for *all* of the blobs in the tracklet is a non-animal label, the tracklet is classified as non-animal, and the most abundant label in the tracklet is chosen as the tracklet label.
3. If *any* of the blobs in the tracklet are classified as multi-animal, the tracklet will be classified as multi-animal. This step will occur only if a multi-animal class has been included in the classifier.
4. If there is no ID label in the sequence of most likely labels, the tracklet is marked as unidentified.
5. Otherwise (i.e. the tracklet is single-animal and there are at least some blobs classified as a specific ID label), we define a score for each possible ID as the sum of the likelihoods for that ID over all blobs:

$$s_l = \sum_k \mathcal{L}_{kl} \tag{7}$$

The tracklet is labeled with the ID that has the maximal score:

$$L = argmax_l(s_l) \tag{8}$$

Where the argmax operation is performed over the labels that represent specific IDs (i.e. excluding the unknown, multi-animal, and no-animal classes). In addition, we define and register the classification confidence score as:

$$S = \frac{n \cdot s_L}{\sum s_l} \tag{9}$$

where n is the number of blobs in the tracklet classified as specific animal IDs, $s_L$ is the score of the assigned label, and the sum is over all label indices that belong to a specific ID (i.e. excluding non-specific labels such as 'Unknown' or 'NoAnimal'). This heuristic score definition takes into account the likelihoods the classifier has assigned to each label, but also the number of *identifiable* blobs in the tracklet. Using this definition, the confidence score will increase as evidence for the assignment accumulates (so longer tracklets with more identifiable blobs will have a higher score).

## 2.5 Verification and retraining

Although this is not the final tracklet ID assignment, it is useful to be able to estimate the performance of tracklet classification. Especially, it is important to assess the performance of a classifier trained on examples from one experiment on tracklets from another. If there is a significant drop in performance, examples from the new experiment can be added to the training set, and an incremental training can be run. Both validation and adding new examples can be done using the anTraX GUI.

## 3. Propagating IDs on the tracklet graph

At this stage, we have the tracklet graph, in which a subset of single-animal tracklets have been labeled with a specific ID and a confidence score for that label. The rest of the tracklets in the network are either unidentified single-animal tracklets, or multi-animal tracklets. We assume that some of these classifications can be incorrect. The next step in the algorithm is to make the actual ID assignments for the tracklets, and to propagate these assigned IDs over the tracklet graph, trying to identify the composition of all tracklets, including multi-animal tracklets. In the process, a large portion of the incorrect classifications will be identified and overridden by the algorithm.

### 3.1 Initializing the graph

We start the propagation algorithm by creating a dynamic list of possible IDs (initially representing all individuals in the experiment) and a dynamic list of assigned IDs (initially an empty list) for each node in the graph. These lists are continuously updated during the propagation process. For all nodes (tracklets) that have been assigned a 'non-animal' label in the classification step, we initialize the possible ID list also as an empty one, effectively removing these nodes from the graph.

## 3.2 Propagation and assignment rules

Propagating and assigning IDs are done according to a set of rules executed in a specific order.

For each node to which we want to assign an ID, we do the following:

1. If the ID we want to assign is not on the list of possible IDs for that node, abort.
2. If the node represents a single-animal tracklet (i.e. is in the area range of a single animal as defined by the user AND was not classified as a multi-animal tracklet by the classifier), assign the ID and eliminate all other possible IDs. If it is not a single-animal node, assign the ID without eliminating other IDs.
3. Horizontal propagation (negative): for all other nodes that overlap in time with the currently assigned node, eliminate the ID we just assigned.
4. Vertical propagation (positive): for each parent node of the current node, look if the currently assigned ID is on the list of possible IDs. If there is only one such parent, and it has not already been assigned the ID, assign the ID to that parent. Do the same for child nodes.
5. Topological propagation (positive): a pair of nodes on the graph that constitute a 2-vertex cut set (i.e. cutting the graph at both these nodes creates a disconnected subgraph) and the corresponding disconnected subgraph does not contain any other 0-indegree or 0-outdegree nodes (i.e. there are no animals leaving or exiting the subgraph), are defined as twin nodes. Such a pair will have exactly the same composition of IDs (this is not true in cases where one of the tracklets in the subgraph touches a border of the ROI at a point where animals can exit and enter; these cases are flagged during tracking, and no assignment is made; the special case of open boundary ROIs is discussed below). For each assignment, we also assign the first descendent twin node and the first ancestor twin node (if they exist).

For each node from which we want to eliminate a possible ID, we do the following (see **Appendix 1—figure 8**, **Figure 2B**, **Figure 2—video 1** for illustrated examples):

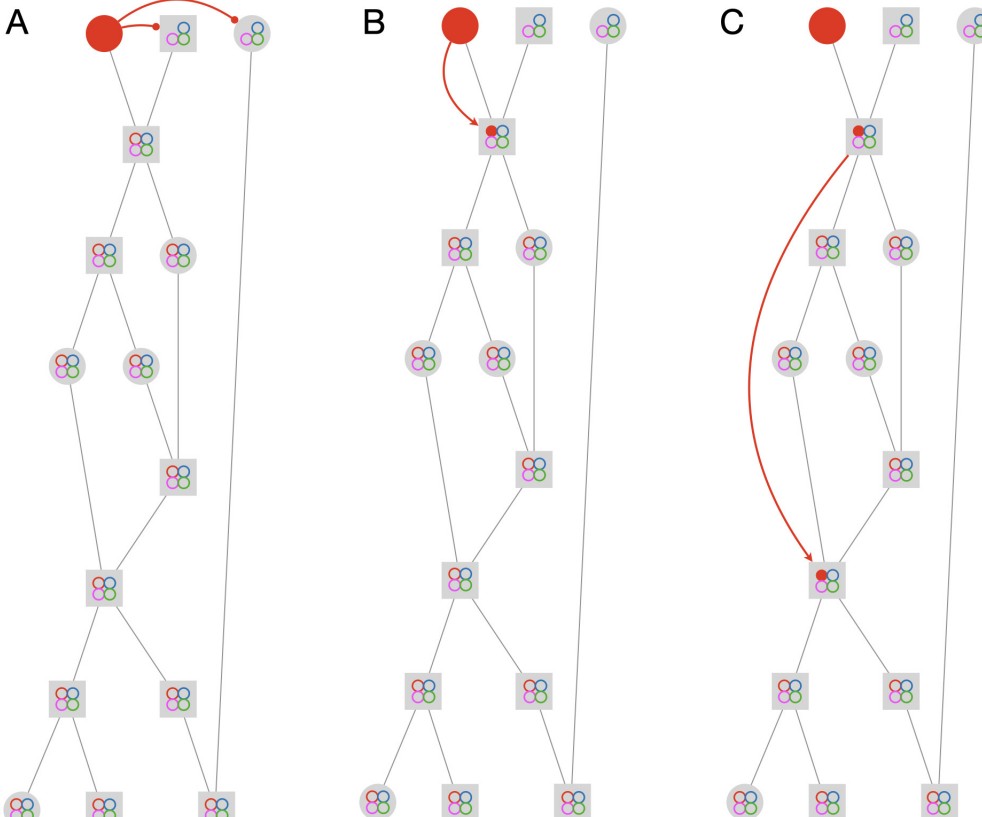

**Appendix 1—figure 8.** Propagation rules. The figure depicts the first three steps in solving an example graph. The graph has 15 tracklets and 4 IDs. Circular nodes mark single-animal tracklets, while square nodes mark multi-animal tracklets. The colored circles inside the nodes mark the

current assignments of the node. Empty circles indicate possible assignments, and full circles indicate actual assignments. The full solution of the example is given in *Figure 2—video 1*. (**A**) Negative horizontal propagation. (**B**) Positive vertical propagation. (**C**) Positive topological propagation.

1. If the ID is already marked as 'assigned' for that node (i.e. the ID was already propagated from that node), abort.
2. Vertical propagation (negative): for each parent node of the current node, if there is no other child node for which the ID that we are currently eliminating is possible, eliminate the ID for that parent. For each child of the current node, if there is no other parent for which the ID is possible, eliminate the ID for that child.
3. Topological propagation (negative): eliminate the ID for the first ancestor twin node and the first descendent twin node (if they exist).

## 3.3 Propagating from classified single-animal tracklets

Before we start the propagation, we rank all the single-animal tracklets that were labeled with a specific ID by the classifier (the 'source' tracklets) according to their confidence scores. We start by assigning the tracklet with the highest score with its classified ID, and then recursively propagate according to the rules above. When no more propagations can be made, we move on to the next tracklet on the list. All nodes with assignments inherit their confidence from the confidence of the source node.

Once the last source tracklet has been reached, we conduct another round of propagation, this time starting from all nodes with assigned IDs (not only the CNN-classified nodes), again sorting them according to their confidence, so that higher confidence propagations will have precedence. This process is repeated until no more propagations can be made.

## 3.4 Handling open boundary ROIs

The assumption that underlies the propagation rules as described above is that a tracklet indeed represents a given set of tagged animals in each of its frames, and that the tracklet graph correctly captures the flow of individual animals between tracklets. This assumption is violated if the ROI of the experiment is open (i.e. animals are free to exit and leave the tracked region), because a tracklet that touches the open boundary can have a changing set of tracked animals. To handle these cases, blobs that overlap with an open boundary are treated differently. In the blob linking step, whenever a blob that touches the open boundary is linked to a blob that does not touch the open boundary in the previous frame, the tracklet closes (even if it is a 1:1 link as defined in section 1.4), and a new tracklet opens and will be linked to the previous with a graph edge. The same happens when a blob that does not touch the boundary is linked to a blob that does. This way, the blobs touching the boundary (i.e. blobs that can 'lose' or 'gain' animals) are confined to the same tracklet. These special tracklets do not participate in the propagation process (i.e. they do not act as sources for IDs and do not accept vertical or topological propagations). Open boundaries are marked by the user as part of the ROI mask creation (see online documentation). See also benchmark dataset V25 (*Figure 3—figure supplement 1*, *Figure 3—video 2*) for an example of a tracked experiment with an open boundary.

## 3.5 Propagation of incorrect classifications

The tracklet classification is never error-free, and some incorrect assignments will be made. As the confidence score of incorrectly classified tracklets will usually be low, the rate of incorrect assignments by the propagation algorithm will usually be lower than the error rate of the classifier. The reason is that, in many cases, these tracklets will already have been assigned by propagation from a more reliable assignment by the time the algorithm reaches them. Nevertheless, some propagation from these incorrectly classified tracklets is to be expected. This propagation will continue until it contradicts an already assigned tracklet. These erroneous propagations are typically short (*Figure 4—figure supplement 2*), and can often be filtered out algorithmically (see next section).

## 3.6 Connected component filtering

Ideally, at this point, when all possible ID propagation options are exhausted, we have inferred the maximum information about the composition of each tracklet. If we look at the subgraph corresponding to a specific ID (defined by all the nodes that are possible for that ID, along with their connecting edges), we expect to see a single connected component (*Appendix 1—figure 9A*). This connected component will consist of nodes assigned with that ID, which do not have nodes parallel to them in the subgraph, as well as nodes without ID assignments, which can in principle have ambiguities (parallel nodes that are members of the same subgraph). However, as discussed in the previous section, because the tracklet classification process usually produces some errors, the ID subgraph can have several disconnected components (*Appendix 1—figure 9B*). To filter out connected components that correspond to classification errors, we assign a confidence score to each connected component, defined as the sum of the confidence scores of all the ID assignments in that component. We then go over the list of components sorted by their confidence, and accept them in order. Whenever a component contradicts one of the already accepted components (e.g. it overlaps in time, or does not contain a possible route on the graph to a previously accepted component), we discard it. To eliminate a component, we undo all assignments made of the focal ID to the nodes of that component, and all the eliminations that resulted from these assignments. This is done separately for each ID subgraph.

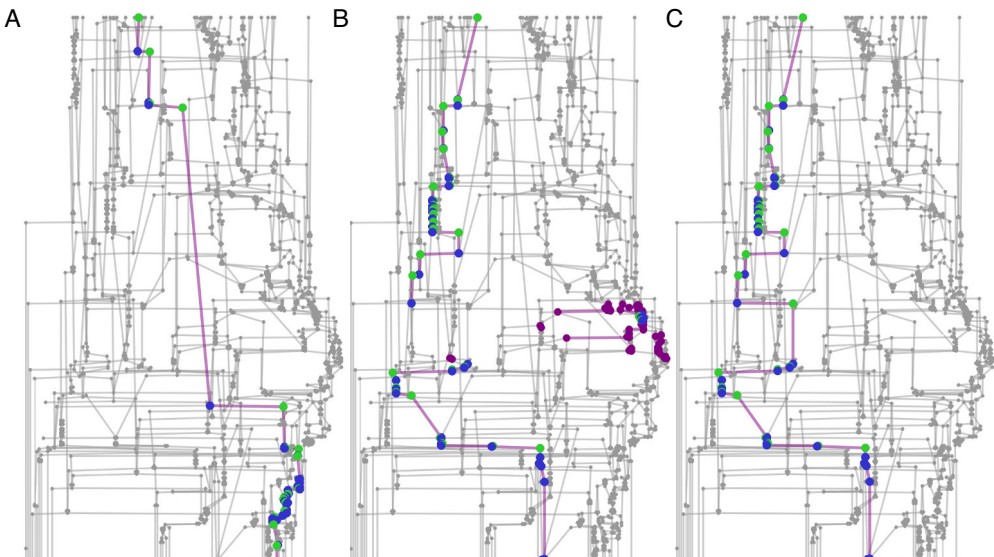

**Appendix 1—figure 9.** Connected component filtering. An example from a 10-min tracklet graph. Green nodes are those assigned by the classifier, blue nodes are assigned by the propagation algorithm, and purple nodes are ambiguous ('possible' but not 'assigned'). (**A**) A focal ant subgraph in which graph assignment propagation was consistent and did not result in contradictions. (**B**) A subgraph for a different focal ant in the same graph, for which the classifier made an incorrect assignment. As a consequence, the subgraph is fractured into a few connected components. (**C**) The subgraph of the same focal ant as in **B**, following the connected component filtering step and a second round of assignment propagations. The erroneous component was filtered, and the algorithm was able to complete the ID path through the graph.

Following the connected component filtering, we again run the ID propagation loop to close the gaps between the accepted components (*Appendix 1—figure 9C*). This procedure is repeated until no more component filtering can be made.

## 3.7 Finalizing assignments and exporting data

At this point, when all inference options are exhausted, each ID is represented in several types of nodes/tracklets. In order of decreasing assignment quality, these are:

1. Single-animal tracklets that were assigned by the classifier and confirmed by the graph propagation algorithm (i.e. that were not identified as erroneous and overridden).
2. Single-animal tracklets for which IDs were inferred by the propagation algorithm.
3. Multi-animal tracklets for which IDs were inferred by the propagation algorithm.
4. Tracklets for which no ID was assigned, but which are the only possible tracklet for a particular ID.
5. Points of ambiguity, where no assignment was made with confidence, and several temporally overlapping nodes could possibly contain the focal ID.

When exporting trajectory data for the experiment, the assignment type for each point in the trajectory is also reported.

## 3.8 Multi-colony experiments

anTraX enables tracking multiple colonies/groups within the same video. This feature is useful when designing and performing high-throughput experiments, where one camera records several colonies. For multi-colony experiments, the software assigns a colony ID to each tracklet during the initial tracking step, based on the spatial location of the tracklet. During the graph propagation step, the software partitions the tracklets into a number of graphs, one for each colony. Propagation is then performed on each colony-graph separately, and the final trajectories are saved separately for each colony. Dataset G6 × 16 (*Figure 3—figure supplement 8*, *Figure 3—video 9*) gives an example of tracking an experiment where 6 colonies of 16 ants each are recorded with a single camera.

