## [Decision Letter]

**Acceptance summary:**

Here, the authors present a new software package for tracking color-tagged animals, anTraX. The method combines object segmentation and tracking through thresholded connected components (blobs), a convolutional neural network that uses shape and color in identify animals when they are alone outside of groups, and then a graph-based method for connecting single animal tracks and group blobs appropriately. This is an interesting and novel strategy that combines aspects of traditional computer vision with newer work using neural networks to allow tracking of tagged animals.

**Decision letter after peer review:**

Thank you for submitting your article "anTraX: high throughput video tracking of color-tagged insects" for consideration by *eLife*. Your article has been reviewed by three peer reviewers, and the evaluation has been overseen by a Reviewing Editor and Catherine Dulac as the Senior Editor. The following individuals involved in review of your submission have agreed to reveal their identity: Joshua W Shaevitz (Reviewer #2); Alfonso Perez-Escudero (Reviewer #3).

The reviewers have discussed the reviews with one another and the Reviewing Editor has drafted this decision to help you prepare a revised submission.

Essential revisions:

As evidenced from the attached reviews, the reviewers were generally positive about the submission, finding the work timely, of significant potential impact, and clearly written. There were several shared concerns, however, that need to be addressed before accepting the article, though.

1) The reviewers all agreed that a key missing feature of the submission was comparing the accuracy, speed, and required computational resources of the method to other existing approaches (e.g., idtracker.ai, DeepLabCut, SLEAP). An analysis along the lines performed in Sridhar et al., 2019 would greatly benefit readers and potential users, making them aware of the apparent benefits and potential disadvantages of the approach described in this submission. If such comparisons are not possible due to the technical limitations of the software, then the authors should clearly describe what the technical limitations of the existing software are and why they could not successfully track their video data.

2) The reviewers also would like to see more description of the limitations of the blob detection algorithm used here. What if the video is mostly made of connected groups (potentially morphing into/out of each other) and there are very few instances of single animals moving about between groups? In experiments at high density, this is certainly the case and other animals that live in groups spend a vast amount of time huddled together. Would anTraX be appropriate for this kind of data? How is the position of each final track found if there is a group involved? Is it just the centroid of the multi-animal blob? Doesn't this cause discontinuities in the final tracks that are problematic for further analysis (e.g., using JAABA as the authors highlight)? For example, socially housed rodents often spend much of their time clustered together. How would anTraX fair on this type of data?

3) The number of ground truth testing frames (200 annotations per data set?) seems rather small. While high-quality ground truth data are difficult to collect, the method the authors use for validating the accuracy of their algorithm via manual labeling as "correct" or "incorrect" (or "skip") could be strongly influenced by sampling bias, or observer bias and fatigue, as is the case with any manual annotation task. This is especially worrying with the small number of annotations used here (although the exact number needs to be spelled out more clearly). We ask that the authors expand their ground truth testing set (or justify why an expansion is not necessary) and to more clearly describe how many testing frames were used and how these frames where chosen.

4) Relatedly, performing an analysis of the method's accuracy as a function of the number of training frames is an important means to assess the practicability of the method to potential users.

5) The anTraX algorithm also uses multiple types of tracking subroutines, so it would be prudent to compare accuracy across these different stages of tracking (e.g. images labeled by the CNN classifier might be very accurate compared to other parts of the tracking). Using only 200 random samples could easily be oversampling frames where individuals are well separated or accurately labeled by the CNN classifier, which would heavily bias the results to look as if the algorithm is highly accurate when the quality of the tracking could in fact be highly variable. Also, for example, it is unclear how a wrong ID propagates across tracklets. While the methods appear robust to the issue, it should be discussed explicitly and addressed here. Performing these types of comparisons across stages of tracking would be informative to the reader to assess whether the approach would be a good one for their particular data.

6) The reviewers felt that the text fails to acknowledge its similarity with previous methods in two aspects. For instance, the concept of tracklet has been used before, at least in Pérez-Escudero et al., 2014 and Romero-Ferrero et al., 2019, with essentially identical criteria to define each tracklet (although this paper presents a significant improvement in the method to assign blobs via optic flow), and the concept of representing the tracking with a network has been used at least in M. Schiegg, P. Hanslovsky, B. X. Kausler, L. Hufnagel, F. A. Hamprecht. Conservation Tracking. Proceedings of the IEEE International Conference on Computer Vision (ICCV 2013), 2013. We accordingly ask the authors to add a more thorough exposition of how anTraX fits into the previous tracking literature.

Reviewer #1:

This paper is an important and timely contribution to the fields of collective, social, and quantitative animal behavior and will certainly serve to make innovative research questions in these fields more tractable, especially those related to group-living animals such as eusocial insects. Overall the paper is well-written and provides a readable review describing the current state-of-the-art for multi-object tracking of animal groups. As the paper makes clear, using computer vision to reliably track groups of individuals over extended periods of time is an exceptionally difficult problem. The authors identify key technical problems with existing software related to disambiguating multiple closely-interacting individuals when tracking groups of animals over long periods. They then provide a clever and comprehensive, albeit complex, solution to address these problems. Their software combines conventional computer vision algorithms with modern deep-learning-based computer vision algorithms and a novel graph-based propagation algorithm that together allow for reliably tracking groups of animals for extended periods of time. The algorithm allows for high-throughput data collection using minimally invasive, human-readable, color-based markers that negate the need for invasive tagging and expensive high-resolution imaging, while also maintaining individual identities through challenging occlusions and ambiguity (even in cases where a human would struggle to disambiguate identities). The code for their software is well written and documented, which must have required an incredible amount of work for which the authors should be commended. They also give clear and useful demonstrations of how to integrate their software with the currently-available suite of tools for quantitative analysis of animal behavior (DeepLabCut, JAABA, etc.), which nicely places their work into the broader context of modern methods for quantifying and analyzing animal behavior.

Despite these many strengths, I have identified a few closely-related issues with the validation methods used to benchmark the anTraX tracking algorithm that, if properly addressed, would strengthen the results presented in the paper:

– No comparisons to existing tracking methods.

– Sample sizes used for assessing accuracy are small with no comparisons of accuracy across time, group size, tag complexity, before/during/after occlusions, or tracking subroutine.

– No direct comparison to the ground-truth trajectory data or "gold standard" barcode tracking.

– No discussion of computation time or complexity.

If the authors are able to address my concerns at least partially, then the manuscript is potentially adequate for publication as a Tools and Resources article but could certainly benefit from the additional analyses described below, especially in regard to my concerns about sample size and sampling bias.

This paper appears to fall into the category described by *eLife* as: "substantial improvements and extensions of existing technologies" in which case "the new method should be properly compared and benchmarked against existing methods", yet the authors make no attempt to compare their software to existing software. The most common form of computer vision-based tracking used in labs today is still basic background subtraction and blob detection followed by some form of energy minimization algorithm such as the linear assignment algorithm (recently reviewed by Sridhar et al., 2019). While the anTraX authors provide comparisons of their software both with and without their novel graph propagation algorithm, their initial tracking algorithm appears to utilize a complex set of ad-hoc criteria for switching between computationally-expensive optic-flow and cheaper object tracking algorithms. Therefore these results are not directly comparable to other methods.

For the uninitiated reader, it would be useful to expand on this discussion in the main text and more clearly demonstrate how the anTraX algorithm compares to commonly-used baseline tracking. The authors appear to have already benchmarked this type of tracking, given their statement “Flow is computationally expensive, we found it to be significantly more accurate than alternatives such as simple overlap or linear assignment (based either on simple spatial distance or on distance in some feature space)”, but do not provide the results of this comparison. In this same vein, it would also be useful to provide direct comparisons to the output of other existing, more complex tracking software (similar to the comparisons made in Sridhar et al., 2019) both quantitatively and qualitatively. A comparison with the recent deep-learning-based idtracker.ai software (Romero-Ferrero et al., 2019), in terms of how quickly identity assignment errors propagate in time between the two algorithms, would be especially relevant. If these comparisons are not possible due to technical limitations of the software, then the authors should clearly describe what the technical limitations of the existing software are and why they could not successfully track their video data. The addition of these comparisons would serve to greatly improve the paper by giving the reader a better idea of how well the anTraX software addresses the problems it claims to solve when compared to a simple baseline as well as more complex existing software.

Second, while I can appreciate that high-quality ground truth data are difficult to collect, the method the authors use for validating the accuracy of their algorithm via manual labeling as "correct" or "incorrect" (or "skip") could be strongly influenced by sampling bias, or observer bias and fatigue, as is the case with any manual annotation task. Relatedly, and perhaps more importantly, it appears that the size of ground truth data used for assessing tracking accuracy is rather small (200 annotations per dataset?), though it is actually not clear to me exactly how much ground-truth data they collected for their analysis. In addition to reporting their sample sizes in the main text, the authors should report this in the transparent reporting form as well, instead of erroneously claiming "no statistical analyses were performed". It's also not clear how these annotations were sampled within the video time series. This is an important point because as the authors are aware, unrecoverable errors tend to propagate as the time series progresses. It would also be useful to make this comparison across time with other tracking algorithms to assess how assignment errors propagate and/or corrected (i.e. by the CNN classifier). Additionally the authors have benchmarked several datasets with varying group size and tag complexity. It would be useful to explicitly analyze (or at least plot) and discuss how these aspects do affect or might affect the quality of the tracking data produced by anTraX (e.g. what are the limits of group-size and tag complexity). The anTraX algorithm also appears to be able to detect when animals come in close contact, so assessing accuracy before, during, and after prolonged occlusions would be very relevant, as this is the main novelty of the paper. The anTraX algorithm also uses multiple types of tracking subroutines, so it would be prudent to compare accuracy across these different stages of tracking (e.g. I would expect images labeled by the CNN classifier to be very accurate compared to other parts of the tracking). Using only 200 random samples could easily be over sampling frames where individuals are well separated or accurately labeled by the CNN classifier, which would heavily bias the results to look as if the algorithm is highly accurate when the quality of the tracking could in fact be highly variable.

Third, comparing the anTraX algorithm to an automated ground-truth would greatly strengthen the results presented in the paper. Using a dataset of animals marked with machine-readable (though not necessarily human-readable) barcodes would be an ideal option for improving this comparison (as well as my first two issues of comparing to other tracking software and improving ground-truth comparisons), as I assume anTraX works with any marked animals, not just color-based markers. The authors actually refer to barcode tracking as the "gold standard", so I was surprised that they did not make any attempt to directly compare the two. I have access to such a barcode dataset (tagged groups of desert locusts, *S. gregaria*, in a laboratory environment) with high-quality ground-truth barcode trajectories that include complex occlusions and ambiguities. I am happy to make part of this dataset publicly available (e.g. on Zenodo) for the authors to freely use for making such a comparison. To circumvent the need for manual labor, the ground truth ID labels from the barcode tracking software could easily be substituted to train the blob classifier CNN with the assumption that a human could, with enough time, disambiguate the barcodes. A second option would be to apply color thresholding to their single tag data (where individuals are marked with a single color), which should give reliable ground-truth trajectories for at least some of the animals (see for example Jolles et al., 2017, Curr. Biol.). A third option would be to use a dataset of solitary animals moving in an arena that can be reliably tracked and then artificially superimposed (using background subtraction) into a combined "multi-individual" video. This would, of course, be less realistic than real data, as ambiguous "interactions" between individuals are less likely to occur, but would be a significant step toward addressing this issue of more comprehensively comparing the tracking data output to a reliable ground truth.

Finally, the authors mention that a high-powered computer or cluster is advised in some cases, so it would be useful to discuss how much processing time is required for running the anTraX algorithm on the different datasets compared with other algorithms.

Comparing the anTraX algorithm across different factors (time, group size, tag complexity, occlusion, subroutine, etc.) with both the simple baseline of blob-detection-based tracking, existing (more complex) software, and the so-called "gold standard" of barcode tracking would go a long way to help better place this work in the broader context of existing tracking algorithms for animal groups. Additionally these added comparisons would help readers make a better informed decision whether or not using the anTraX software (vs. other tracking software) is worth the investment for their research project.

Reviewer #2:

The authors present a new software package for tracking color-tagged animals, anTraX. The method combines object segmentation and tracking through thresholded connected components (blobs), a CNN that uses shape and color in identify animals when they are alone outside of groups, and then a graph-based method for connecting single animal tracks and group blobs appropriately. This is an interesting strategy that combines aspects of traditional computer vision with newer work using NNs to allow tracking of tagged animals.

In general, I think this is an interesting technique that occupies the space somewhere between traditional tracking methods and new deep-learning approaches to tracking animals and their body parts. I have a few concerns that should be addressed before publication.

1) My biggest worry with this technique is about the treatment of the animal groups (the multi-animal blobs).

– What if the video is mostly made of connected groups (potentially morphing into/out of each other) and there are very few instances of single animals moving about between groups? In experiments at high density this is certainly the case and other animals that live in groups spend a vast amount of time huddled together. Would anTraX be appropriate for this kind of data? I worry that because of this issue anTraX is less broadly applicable than pitched in the article.

– How is the position of each final track found if there is a group involved? Is it just the centroid of the multi-animal blob? Doesn't this cause discontinuities in the final tracks that are problematic for further analysis (e.g. using JAABA as the authors highlight)? Also, JAABA doesn't really apply for animals when they are in a group, right, because the resultant track is shared among all animals in the group?

– The authors highlight a DLC analysis of the single animal images. But this fails for groups, right? I think this needs to be said more clearly.

2) At several steps in the analysis the segments (blobs) are fit to ellipses. This obviously makes sense for elongated objects, but what if the animals segments (which are inherently smoother than the animal itself) are essentially round? Would this affect the analysis?

3) How are the training images for the CNN found? The text says "To train the classifier, we collect a set of example images for each classifier label (Appendix—figure 4-6). This can be done easily using an anTraX GUI app (see Online Documentation for details).". The authors should describe how the training images are selected etc in the main manuscript. Are they taken from a random sample of the single animal blobs? What about animals that are rarely represented in that set, e.g. if an animal is almost always in the group but rarely appears as a single?

4) Recent advances in NN-based pose tracking now allow for multiple animals (see maDLC and SLEAP on which I am an author). I realize that these packages just recently became available but it would be useful for the authors to compare their method to those which don't utilize the tags for ID. This is not strictly necessary for a revision but would clearly be of interest to the field.

Reviewer #3:

The paper presents a tracking system for manually marked individuals. Overall, I think it's a really good paper, and the software seems a useful contribution, both for potential users and for future development of similar tracking systems. The paper is clear and well written, and the conclusions seem well supported by the results. All my issues are mostly about presentation:

1) One of the most time-consuming steps is training of the classifier, in which the user must manually annotate many images of the animals. This step is almost absent from the current main text. Even the supplement does not give an estimate of how many images need to be manually annotated. In my opinion, the main text should explicitly address this step, and include an estimate of the amount of manual work needed.

2) Error propagation may be an issue with this algorithm, since a wrong ID can propagate across tracklets. The method seems quite robust against this issue, but I think that it should be discussed explicitly and addressed in the validation. To do so, I think the validation should include more information. For all the mistakes detected, it should report length of the tracklet with wrong ID and the certainty of the ID assignment (since this certainty will correlate with the probability of propagation). Also, for each mistake detected, the authors should check whether that mistake propagated to any neighboring tracklets. While this must be done manually, given the low number of mistakes after ID propagation, it should be easily doable.

3) In my opinion the method has enough novelty to grant publication in *eLife*, but I feel that the text fails to acknowledge its similarity with previous methods in two aspects: First, the concept of tracklet has been used before, at least in Pérez-Escudero et al., 2014 and Romero-Ferrero et al., 2019, with essentially identical criteria to define each tracklet (although this paper presents significant improvement in the method to assign blobs via optic flow). Second, the concept of representing the tracking with a network has been used at least in M. Schiegg, P. Hanslovsky, B. X. Kausler, L. Hufnagel, F. A. Hamprecht. Conservation Tracking. Proceedings of the IEEE International Conference on Computer Vision (ICCV 2013), 2013.

4) I find that one of the main limitations to usability of this software is that installation seems hard. From the description in the web, I understand that I need all the following: (1) A Linux machine, (2) Python (and know what a virtual environment is and how to create it), (3) Matlab or Matlab's MCR, (4) git (and knowledge about how to clone a repository). Given that the target audience of this software are experimentalists in the field of animal behavior, I think that these requirements will severely damage its usability. And it seems that part of these requirements are in fact unnecessary (at least git should not be needed for basic users). And even having the necessary skills, the need of so many different steps makes me worry that something will fail along the way. I think that a modest investment in simplifying this step will increase the number of users substantially.

---

## [Author Response]

Essential revisions:As evidenced from the attached reviews, the reviewers were generally positive about the submission, finding the work timely, of significant potential impact, and clearly written. There were several shared concerns, however, that need to be addressed before accepting the article, though.1) The reviewers all agreed that a key missing feature of the submission was comparing the accuracy, speed, and required computational resources of the method to other existing approaches (e.g., idtracker.ai, DeepLabCut, SLEAP). An analysis along the lines performed in Sridhar et al., 2019 would greatly benefit readers and potential users, making them aware of the apparent benefits and potential disadvantages of the approach described in this submission. If such comparisons are not possible due to the technical limitations of the software, then the authors should clearly describe what the technical limitations of the existing software are and why they could not successfully track their video data.

The issue with comparing anTraX to other software packages is that there is no other software that can directly track the types of experiments we developed anTraX for. This includes the benchmark experiments presented in this manuscript. Below we briefly review the existing tracking software and how they fall short in tracking these kinds of videos successfully.

General marker-less multi-object tracking algorithms: Obviously, color tagged animals can be tracked as if they were marker-less by general-purpose multi-object tracking algorithms. However, these are rarely useful when it is important to keep track of the identity of each individual animal throughout the experiment. As we review in the Introduction of our paper, most of these approaches can overcome simple occlusions, crossings, and interactions by either trying to segment animals from each other (requiring sufficient image quality and resolution) or using various kinematic modeling approaches to connect tracks following an occlusion to the tracks preceding it. These methods work only for brief, simple interactions, and will fail on experiments like ours, which sacrifice resolution and image quality for duration and high throughput and focus on social insects that tend to cluster tightly together for long periods. It’s also important to note that even a relatively low identity switching error (say 1%) entails the complete mixing of identities in experiments that are much longer than the typical time between interactions. Nevertheless, marker-less methods are still heavily used in our field, and are usually combined with extensive manual correction steps to overcome these issues.

Multi animal DLC and SLEAP: While these newly introduced methods are primarily designed for pose tracking, they can also be viewed as pose-assisted centroid trackers, in which pose information is used to solve cases of overlap and occlusion. From the point of view of multiple object tracking, these methods can be considered sophisticated marker-less trackers, as they do not attempt to classify or recognize the individual animals, but rather to relate objects in each frame in the video to their location in the previous frame. Several disadvantages make these algorithms unsuitable for tracking our experiments. First, they require high image resolution to allow reliable pose tracking of individual animals. Second, as they use the same model to pose-track all the animals, they will not handle experiments of groups with morphological variability well (e.g. our C12 dataset). Third, like the more traditional marker-less trackers mentioned above, they will fail on tight aggregations where even the human eye cannot identify single animals and will suffer from accumulation of identity switch errors in any case of switch error probability in crossings that is not strictly zero.

idTracker.ai: This is the only software currently available that is theoretically capable of tracking color tagged insects, as the reviewers correctly mention below. Its use of neural networks to learn a signature of the tracked animals has the potential to overcome the accumulation of switch errors that happens in marker-less trackers. However, there are several issues that prevent idTracker.ai from working on our data “as is”. First, idTracker.ai collects training sets by looking for points in the videos where all animals are segmented individually (i.e., the number of blobs equals the number of animals). This does not occur in most of our datasets, and also not in many typical experiments involving social insects. Second, idTracket.ai currently works by converting the frames to greyscale, thus discarding most of the relevant information for identification in our videos (although it does surprisingly well on short videos with the greyscale signature of the color tags). While this could be easily solved by a simple modification to the algorithm, doing so is beyond the scope of benchmark comparisons. Third, the idTracker.ai algorithm itself makes it impractical to track videos longer than 30 minutes, as the execution time climbs strongly supra-linearly with the length of the video. Because of this, the types of experiments we optimized anTraX for really fall outside of the current scope of idTracker.ai, and we therefore prefer not to include a formal performance comparison between idTracker.ai and anTraX in the paper.

In summary, we feel that our discussion of existing tracking software in the Introduction is sufficient, and that a formal comparison would not be any more informative.

2) The reviewers also would like to see more description of the limitations of the blob detection algorithm used here. What if the video is mostly made of connected groups (potentially morphing into/out of each other) and there are very few instances of single animals moving about between groups? In experiments at high density, this is certainly the case and other animals that live in groups spend a vast amount of time huddled together. Would anTraX be appropriate for this kind of data? How is the position of each final track found if there is a group involved? Is it just the centroid of the multi-animal blob? Doesn't this cause discontinuities in the final tracks that are problematic for further analysis (e.g., using JAABA as the authors highlight)? For example, socially housed rodents often spend much of their time clustered together. How would anTraX fair on this type of data?

The reviewers are correct: in order to work, anTraX assumes a mixture of single-animal and multi-animal tracklets. In that sense, it is not a universal tool. If a video includes only multi-animal tracklets, anTraX won’t have anything to work with, and will fail to track the animals. However, the case of a mixture of multi-ant tracklets and single-ant tracklets represents most behavioral modes in the study of collective behavior of social insects, which is the target user-base of this tool. anTraX deals well with scenarios in which animals “spend a vast amount of time huddled together”, as long as individuals can be segmented and identified occasionally, which will be the case in almost all biologically realistic scenarios.

We are now explicitly mentioning this requirement in the text:

“The successful propagation of individual IDs on top of the tracklet graph requires at least one identification of each ID at this step. Propagation will improve with additional independent identifications of individuals throughout the video.”

In the context of this answer, it is also useful to note that the necessity of segmenting each animal individually at some point during the experiment is much relaxed compared to the requirement of the current state-of-the-art (idTracker.ai) to have all animals segmented individually in the same frame at some point during the experiment.

The location of an individual in a multi-animal blob is assumed to be at the centroid. As we write in the manuscript:

“Locations estimated from multi-animal tracklets are necessarily less accurate than locations from single-animal tracklets, and users should be aware of this when analyzing the data. For example, calculating velocities and orientations is only meaningful for single-animal tracklet data, while spatial fidelity can be estimated based also on approximate locations.”.

This is something to be aware of when analyzing positional data, as generally kinematic measures will not be usable for groups. However, many animal behavior analyses are positional, and not kinematic, so knowing in which group the animal resides is useful information.

anTraX does not simply export trajectory data to JAABA. anTraX implements its own version of the first step of the JAABA analysis pipeline – the per-frame features generation, where the trajectories are projected into a high dimensional feature space – and writes NaN for the kinematic features in those cases. It also defines many other, anTraX specific, features that describe the properties of the tracklets: the number of animals in the group, the group area and location relative to other animals, etc. A full list of these features and their definitions is given in the online documentation. JAABA can then be used to either classify the behavior of the animal when isolated (for that to work, we added a few negative examples from groups, which JAABA then uses to learn this behavior is for isolated individuals only), or to classify behavior in groups that is not dependent on kinematics. We have modified both the paper and the online documentation (https://antrax.readthedocs.io/en/latest/jaaba/) to emphasize this. The revised manuscript now reads:

“Beyond useful information about the appearance and kinematics of the tracked animals, these extra features provide information about whether an animal was segmented individually or was part of a multi-animal blob. This enables JAABA to learn behaviors that can only be assigned to individually segmented animals, such as those that depend on the velocity of the animal.”

3) The number of ground truth testing frames (200 annotations per data set?) seems rather small. While high-quality ground truth data are difficult to collect, the method the authors use for validating the accuracy of their algorithm via manual labeling as "correct" or "incorrect" (or "skip") could be strongly influenced by sampling bias, or observer bias and fatigue, as is the case with any manual annotation task. This is especially worrying with the small number of annotations used here (although the exact number needs to be spelled out more clearly). We ask that the authors expand their ground truth testing set (or justify why an expansion is not necessary) and to more clearly describe how many testing frames were used and how these frames where chosen.

Because concerns regarding our validation method are alluded to in several reviewer comments, we provide an in-depth overview of our rational in choosing this method here, followed by a description of improvements made in response to these comments.

The ideal validation of anTraX, like any tracking algorithm, is against a ground truth, i.e. the “real” trajectories of each individual in the group, for the duration of the entire experiment, and for all benchmark datasets. The common way to generate such ground truth data is by performing manual tracking by a human, or preferably by a number of people independently. No such ground truth data readily exist for experiments that represent the intended use-niche of anTraX: long-term video recordings of groups of color tagged social insects. Moreover, such ground truth data is practically impossible to generate for any of our benchmark datasets (spanning altogether trajectories in total length of thousands of hours).

Therefore, several possibilities can be considered to validate our algorithm:

1) Benchmark anTraX using smaller datasets, for example by taking a short segment of few of our benchmark datasets and manually tracking them fully. While this approach will definitely give the most reliable performance estimation *for the tested dataset*, it will introduce a significant sampling bias. One of the challenges in tracking long experiments is the non-stationarity, expressed in slow transitions between many behavioral modes, changes in background, changes in tag quality, etc. Taking a short segment out of a long experiment will miss this behavioral and experimental complexity and will not necessarily be predictive of how well anTraX does on long experiments.

2) Comparing anTraX to tracking results of an existing algorithm that is assumed to perform close-to-perfectly. However, as we explained above, no such algorithm exists for tracking data of the kind we optimized anTraX for (color-tagged groups of closely interacting insects). Reviewer #1 suggested several clever ways around this problem: First, they suggested using the fact that our approach is actually not limited to color-tags, but can classify individual animals based on any image feature. Such features can include the natural appearance of the animals (not unlike the principle behind idTracker.ai), or the features of a machine-readable barcode attached to the animal, which can be tracked well by existing algorithms. Reviewer #1 also very kindly offered to share a dataset of tracked barcode marked animals. However, although we would be excited to try this approach and thank the reviewer for their generosity, we feel this is more of an “off label” use of the algorithm. We optimized the anTraX image processing pipeline and CNN classifier for use with low-resolution images and color-based information. It would need to be significantly changed in order to work equally well with barcodes. This validation method also suffers from the problem that performance would be estimated against a benchmark outside the range of scenarios we developed anTraX to handle.

3) A second suggestion by reviewer #1 is to compare anTraX’s performance on groups of animals tagged with a single-color mark each, to a simple tracking approach (color blob tracking). The problem with this suggestion is that such a tracker is not expected to perform well on cases of tag occlusion (e.g., when an animal grooms itself) or animal interactions, and hence cannot serve as a gold standard. This method also suffers again from a sampling bias, this time choosing a relatively easy tracking scenario, with a low number of individuals.

4) The reviewer’s third suggestion is to create an “artificial colony” by superimposing a few single animal videos, each with a unique color tag. This way, ground truth data can be generated by tracking the single animal videos, while using anTraX to track the combined video. While this is an interesting suggestion, it suffers again from the problem of not testing anTraX directly on the type of experiments and range of behaviors we designed it for. It could be a nice complementary approach but will not give a good sense of how anTraX performs on “real” data.

We feel that our chosen approach, while not perfect, has significant advantages over these alternatives:

1) The random sampling (of timepoints and individuals) of validation points across the entire experiment offers the most unbiased way of subsampling our extensive benchmark dataset collection. This allows for a simple way to estimate the gross performance of the algorithm over a wide range of experimental conditions.

2) anTraX’s validation interface also allows us to narrow down the range of tracking point selection by time or individual, thus allowing a more specific estimation of tracking performance. For the purpose of this tools-and-resources paper, we think the average performance for a given dataset is the most informative. For analyses more specific to certain behaviors, this feature is most useful. For example, in a soon-to-be-published work (preprint available at https://www.biorxiv.org/content/10.1101/2020.08.20.259614v1), we analyzed the foraging behavior of ant colonies, and estimated the tracking performance during very specific events in time, and for specific ants of interest.

3) The random sampling of validation points, together with the simple binary measure, allows for estimating confidence intervals for the tracking error. This enables a simple determination of the number of points required for validation, based on the accuracy needed for such an estimation.

4) It is true that like any manual annotation task, our method suffers from observer bias and fatigue, as the reviewers pointed out. The best way around this is using multiple annotations of the same dataset. However, this is a classical bias/variance tradeoff: it is the same total effort to annotate N points twice (lower bias but lower confidence) or 2N points once (higher bias but higher confidence). Since our algorithm does not provide marginal improvement over other methods, but rather introduces the ability to track experiments that could not be tracked by any other existing method (as we detail in the answer to issue #1), we felt that optimizing this tradeoff is of less importance.

5) For the same reason, we felt that demonstrating how our algorithm is able to successfully track many conditions is of more relevance than giving very precise error estimations. This is the reason we were originally satisfied with 200 validation points per experiment, which corresponds to confidence intervals of below 1-3%.

In order to address, at least partially, the reviewers’ concerns regarding the validity of our tracking performance estimation, we expanded our analysis in the following ways:

1) We increased the number of total validation points to 500 (from 200 in the original analysis) for each experiment. This significantly reduces the confidence intervals of the estimates (estimates and confidence intervals are reported in Table 2). The 500 points in the new analysis were resampled from the experiments and did not use the 200 points used in the original analysis. Noticeably, the difference between the estimated error in the original analysis and that of the new expanded analysis was negligible, providing further reassurance that our validation approach is reliable.

2) For each of the validation points, we extracted the duration of the tracklet to which it belongs. Under the assumption that validation of a point along a tracklet represents the entire tracklet well, we estimated that the total 9500 validation points used across all datasets represent a total of almost 4 hours of trajectory data. Thus, crudely, our analysis is equivalent to benchmarking against a fully tracked dataset of that volume, albeit sampled randomly across all the experimental conditions represented in our datasets.

3) To address point #5 below, we further analyzed all validation points according to their type of assignment, either directly by the classifier, or by the propagation algorithm. We estimated the assignment accuracy per assignment type. No significant difference was found in the accuracy of the two types of assignments. The results of this analysis are reported in the new Figure 4—figure supplement 1.

4) To address point #2 of reviewer #3, we added an analysis of the distribution of lengths of erroneous segments. While this is not directly correlated with overall performance, it is informative with regard to the algorithm’s ability to block error propagation. This analysis is reported in the new Figure 4—figure supplement 2.

5) We expanded the description of the rational and details of the validation method in the revised version, including a clear statement about the size of the validation set, and the way it was sampled:

“Because the recording duration of these datasets is typically long (many hours), it is impractical to manually annotate them in full. Instead of using fewer or smaller datasets, which would have introduced a sampling bias, we employed a validation approach in which datasets were subsampled in a random and uniform way. In this procedure, a human observer was presented with a sequence of randomly selected test points, where each test point corresponded to a location assignment made by the software to a specific ID in a specific frame.”

“The process was repeated until the user had identified 500 points as either correct or incorrect”.

“This procedure samples the range of experimental conditions and behavioral states represented in each of the datasets in an unbiased manner, and provides a tracking performance estimate that can be applied and compared across experiments.”

6) We have amended the transparent reporting form to include details of the performance estimation to reflect the error estimation process.

We hope that the reviewers will appreciate that this was a significant amount of work, and that they will deem our improved and expanded validation approach suitable for the purpose of this publication.

4) Relatedly, performing an analysis of the method's accuracy as a function of the number of training frames is an important means to assess the practicability of the method to potential users.

Like any supervised machine learning algorithm, the accuracy of the blob classifier in anTraX depends on the size and quality of the training set. While it is true that, generally, the more examples are given to the NN to train on the more accurate it will be, the size of the training set is less important than the distribution of the examples in the image space. A good training set will have denser distribution of examples in regions of the image space that are both harder to classify (i.e., pack closely images that belong to different individuals) and relevant for the task (i.e., images that represent the actual data well). Like many supervised machine learning tracking algorithm (e.g., LEAP, SLEAP, DLC), anTraX works best using an iterative training approach: starting by either using an existing classifier (possibly trained for a different experiment) or one trained on a limited set of easy examples for a first pass tracking run, then using misclassified images to enhance the training set. This recommended workflow is described in detail in the online documentation.

In the revised manuscript, we have addressed the reviewers’ request as follows:

1) We have added an analysis of the accuracy of the “blob classifier” as a function of number of training frames. The training frames have been resampled randomly from the complete training set. The results are reported in the new Figure 4—figure supplement 3.

2) We have also added an analysis of the relationship between the accuracy of the blob classifier and the final performance of the tracking algorithm. This gives the user a sense of what to aim for when iteratively training the blob classifier. This is reported in the new Figure 4—figure supplement 3.

3) In addition, as a response to a request by reviewer #2, we have expanded the detailed description of the training step in the Appendix:

“In short, the GUI presents the user with all the blob images from a random tracklet. The user can then select the appropriate ID and choose to either export all images into the training set, or to select only a subset of images (useful if not all blobs in the tracklet are recognizable). In many cases, especially in social insects where behavioral skew can be considerable, some animals are rarely observed outside an aggregation. It is therefore challenging to collect examples for them using a random sampling approach. One solution to this problem, which is the recommended one for high throughput experiments, is to pool data from several experiments into one classifier as discussed in the main text. Another solution, in case this is not possible, is to scan the video for instances in which the focal animal leaves the group, and “ask” the GUI for tracklets from this segment of the video. Alternatively, one can do a first pass of classification (not full tracking but simply running the blob classifier), and then ask the GUI to display only unclassified tracklets, increasing the probability of spotting the missing animal.”

5) The anTraX algorithm also uses multiple types of tracking subroutines, so it would be prudent to compare accuracy across these different stages of tracking (e.g. images labeled by the CNN classifier might be very accurate compared to other parts of the tracking). Using only 200 random samples could easily be oversampling frames where individuals are well separated or accurately labeled by the CNN classifier, which would heavily bias the results to look as if the algorithm is highly accurate when the quality of the tracking could in fact be highly variable. Also, for example, it is unclear how a wrong ID propagates across tracklets. While the methods appear robust to the issue, it should be discussed explicitly and addressed here. Performing these types of comparisons across stages of tracking would be informative to the reader to assess whether the approach would be a good one for their particular data.

We have performed this analysis, and the results are reported in the new Figure 4—figure supplement 1. See also our response to issue #3 above.

6) The reviewers felt that the text fails to acknowledge its similarity with previous methods in two aspects. For instance, the concept of tracklet has been used before, at least in Pérez-Escudero et al., 2014 and Romero-Ferrero et al., 2019, with essentially identical criteria to define each tracklet (although this paper presents a significant improvement in the method to assign blobs via optic flow), and the concept of representing the tracking with a network has been used at least in M. Schiegg, P. Hanslovsky, B. X. Kausler, L. Hufnagel, F. A. Hamprecht. Conservation Tracking. Proceedings of the IEEE International Conference on Computer Vision (ICCV 2013), 2013. We accordingly ask the authors to add a more thorough exposition of how anTraX fits into the previous tracking literature.

We thank the reviewers for pointing this out. We definitely borrowed the term “tracklet” from previous work and did not mean to make it seem otherwise. We have added the appropriate citation to the revised manuscript:

“First, similar to other multi-object tracking algorithms, it segments the frames into background and ant-containing *blobs and* organizes the extracted blobs into trajectory fragments termed tracklets (Pérez-Escudero et al., 2014, Romero-Ferrero et al., 2019).”

We also appreciate pointing us to the line of work using graph models to represent tracking data and we explicitly mention it in the revised manuscript:

“The tracklets are linked together to form a directed tracklet graph (Nillius et al., 2006)”

“While formal approaches for solving this problem using Bayesian inference have been proposed (Nillius et al., 2006), we chose to implement an ad-hoc greedy iterative process that we found works best in our particular context.”

Reviewer #1:This paper is an important and timely contribution to the fields of collective, social, and quantitative animal behavior and will certainly serve to make innovative research questions in these fields more tractable, especially those related to group-living animals such as eusocial insects.[…]Comparing the anTraX algorithm across different factors (time, group size, tag complexity, occlusion, subroutine, etc.) with both the simple baseline of blob-detection-based tracking, existing (more complex) software, and the so-called "gold standard" of barcode tracking would go a long way to help better place this work in the broader context of existing tracking algorithms for animal groups. Additionally these added comparisons would help readers make a better informed decision whether or not using the anTraX software (vs. other tracking software) is worth the investment for their research project.

We appreciate the reviewer’s many suggestions regarding the validation of the anTraX algorithm, especially their generous offer to share data! We have addressed many of the major points in our response to issue #3 in the combined review. To answer also the other points:

Comparing to baseline trackers: This is indeed of interest to us as developers, and during the design and development of anTraX we have considered and measured the performance of many alternatives to the different computational steps in the “baseline” tracking. However, the anTraX approach here is very conservative and traditional. This is not where the novelty of anTraX lies. Moreover, we do not make any claims to outperform other algorithms here, and as our choices were mostly guided by the needs of later stages in the algorithm, it is reasonable to assume that other algorithms will outperform anTraX. We agree that a systematic comparison between different segmentation and linkage algorithms would be interesting and useful for the community, but feel it is well outside of the scope of the current paper.

Comparing performance across group size and tag complexity: While it is true that our benchmark datasets contain examples for various tag complexities and group sizes, they also vary in many other important properties (e.g., image quality, resolution, species, number of colors). While readers are free to look at the raw numbers reported in Tables 1 and 2, we are hesitant to make explicit claims regarding the dependency of performance on any specific factor. Such an analysis will require a properly controlled experiment in which the feature in question varies while the others are held constant.

Computer clusters: anTraX’s approach to parallelization is straight forward: tracking the experiment in chunks, each tracked as a different job/thread, and running a “stitching” step at the end. The gain is therefore proportional to the parallelization factor used, minus the very low overhead in the stitch step. Because we developed anTraX for high throughput, long duration experiments, and unlike other software packages, we included a built-in interface to handle this parallelization.

Reviewer #2:[…]1) My biggest worry with this technique is about the treatment of the animal groups (the multi-animal blobs).– What if the video is mostly made of connected groups (protentially morphing into/out of each other) and there are very few instances of single animals moving about between groups? In experiments at high density this is certainly the case and other animals that live in groups spend a vast amount of time huddled together. Would anTraX be appropriate for this kind of data? I worry that because of this issue anTraX is less broadly applicable than pitched in the article.– How is the position of each final track found if there is a group involved? Is it just the centroid of the multi-animal blob? Doesn't this cause discontinuities in the final tracks that are problematic for further analysis (e.g. using JAABA as the authors highlight)? Also, JAABA doesn't really apply for animals when they are in a group, right, because the resultant track is shared among all animals in the group?

This has been addressed in the response to issue #2 in the combined review section.

– The authors highlight a DLC analysis of the single animal images. But this fails for groups, right? I think this needs to be said more clearly.

Yes! Pose tracking is currently only available for individually segmented animals. We have emphasized this more clearly in the revised version. We plan to integrate the recently introduced maDLC and SLEAP to extend this into multi-animal tracklets, although tight groups will probably not benefit much from this. We have emphasized this point in the revised version:

“Currently, this is only supported for single-animal tracklets, where animals are well segmented individually.”

We also added a discussion point with regard to multi animal pose estimation:

“Lastly, a newer generation of pose-estimation tools, including SLEAP (Pereira et al., 2020) and the recent release of DeepLabCut with multi-animal support, enable the tracking of body parts for multiple interacting animals in an image. These tools can be combined with anTraX in the future to extend pose tracking to multi-animal tracklets, and to augment positional information for individual animals within aggregations.”

2) At several steps in the analysis the segments (blobs) are fit to ellipses. This obviously makes sense for elongated objects, but what if the animals segments (which are inherently smoother than the animal itself) are essentially round? Would this affect the analysis?

The main use of the ellipse fitting in the algorithm is to assign an initial orientation to the blob. This orientation is not used in the algorithm itself; it is just added to the tracking output. For low eccentricity blobs animals, this will be practically meaningless, but will not affect the tracking algorithm. If orientation is needed for such cases, it can be also found using pose tracking, or using the blob classifier (for identifiable blobs only).

3) How are the training images for the CNN found? The text says "To train the classifier, we collect a set of example images for each classifier label (Appendix—figure 4-6). This can be done easily using an anTraX GUI app (see Online Documentation for details).". The authors should describe how the training images are selected etc in the main manuscript. Are they taken from a random sample of the single animal blobs? What about animals that are rarely represented in that set, e.g. if an animal is almost always in the group but rarely appears as a single?

We have expanded the technical description of the training set collection in the appendix to include these details:

“In short, the GUI presents the user with all the blob images from a random tracklet. The user can then select the appropriate ID and choose to either export all images into the training set, or to select only a subset of images (useful if not all blobs in the tracklet are recognizable). In many cases, especially in social insects where behavioral skew can be considerable, some animals are rarely observed outside an aggregation. It is therefore challenging to collect examples for them using a random sampling approach. One solution to this problem, which is the recommended one for high throughput experiments, is to pool data from several experiments into one classifier as discussed in the main text. Another solution, in case this is not possible, is to scan the video for instances in which the focal animal leaves the group, and “ask” the GUI for tracklets from this segment of the video. Alternatively, one can do a first pass of classification (not full tracking but simply running the blob classifier), and then ask the GUI to display only unclassified tracklets, increasing the probability of spotting the missing animal.”

4) Recent advances in NN-based pose tracking now allow for multiple animals (see maDLC and SLEAP on which I am an author). I realize that these packages just recently became available but it would be useful for the authors to compare their method to those which don't utilize the tags for ID. This is not strictly necessary for a revision but would clearly be of interest to the field.

SLEAP and maDLC are definitely exciting developments in the field, and we have been waiting for their publication. While these methods are primarily for pose-tracking, they can also be viewed as pose-assisted centroid trackers, where pose information is used to solve cases of overlap and occlusions. We did not refer to these methods in the original manuscript because it was submitted a couple of weeks before their release. The revised version now explicitly describes and cites them.

In short, and as we described earlier in this response letter, these methods can be seen as sophisticated marker-less trackers. As such, they suffer from the same disadvantages as other marker-less trackers: accumulation of identity switching errors and dependency on high image quality. Therefore, they are less suited to directly track the same types of experiments as anTraX.

We see these methods as complementary to anTraX, and plan on porting them into the anTraX interface in a similar manner to what we did for single animal DLC. This will allow us to extend pose tracking, which, as the reviewer highlighted above, works currently only for single animal tracklets, to multi animal tracklets. We also hope to use their advantage in separating overlapping animals to improve our tracking accuracy by better locating individuals in multi animal blobs. The revised version now expands on this point in the Discussion:

“Lastly, a newer generation of pose-estimation tools, including SLEAP (Pereira et al., 2020) and the recent release of DeepLabCut with multi-animal support, enable the tracking of body parts for multiple interacting animals in an image. These tools can be combined with anTraX in the future to extend pose tracking to multi-animal tracklets, and to augment positional information for individual animals within aggregations.”

Reviewer #3:The paper presents a tracking system for manually marked individuals. Overall, I think it's a really good paper, and the software seems a useful contribution, both for potential users and for future development of similar tracking systems. The paper is clear and well written, and the conclusions seem well supported by the results. All my issues are mostly about presentation:1) One of the most time-consuming steps is training of the classifier, in which the user must manually annotate many images of the animals. This step is almost absent from the current main text. Even the supplement does not give an estimate of how many images need to be manually annotated. In my opinion, the main text should explicitly address this step, and include an estimate of the amount of manual work needed.

As we describe in our response to issue #4 in the combined review, we have added an analysis of the number of frames required to train the blob classifier. We also expanded the description of the labeling procedure in the appendix. Note, however, that as the labeling is done per tracklet (which can have up to a few hundred blobs in some experiments), the number of labeled blobs is always a good predictor of the manual work needed.

2) Error propagation may be an issue with this algorithm, since a wrong ID can propagate across tracklets. The method seems quite robust against this issue, but I think that it should be discussed explicitly and addressed in the validation. To do so, I think the validation should include more information. For all the mistakes detected, it should report length of the tracklet with wrong ID and the certainty of the ID assignment (since this certainty will correlate with the probability of propagation). Also, for each mistake detected, the authors should check whether that mistake propagated to any neighboring tracklets. While this must be done manually, given the low number of mistakes after ID propagation, it should be easily doable.

As the reviewer mentions, an incorrect classification will indeed propagate locally to multi-ant tracklets and unidentified single ant tracklets. However, in the majority of cases, such propagations will not be carried for long, and will terminate due to a contradiction, either by bumping into a correct classification of the individual, or by contradicting a higher-ranked assignment of the incorrect ID. Assuming the error rate is not too high, this will result in a small disconnected ID subgraph (see Appendix section “Propagating IDs on the tracklet graph”, subsection “Propagation of incorrect classifications”, for definition), that will be filtered out (Appendix—figure 9).

We have included an analysis in line with what the reviewer suggested, reported in the new Figure 4—figure supplement 2 (see also our response to issue #3 in the combined review).

3) In my opinion the method has enough novelty to grant publication in eLife, but I feel that the text fails to acknowledge its similarity with previous methods in two aspects: First, the concept of tracklet has been used before, at least in Pérez-Escudero et al., 2014 and Romero-Ferrero et al., 2019, with essentially identical criteria to define each tracklet (although this paper presents significant improvement in the method to assign blobs via optic flow). Second, the concept of representing the tracking with a network has been used at least in M. Schiegg, P. Hanslovsky, B. X. Kausler, L. Hufnagel, F. A. Hamprecht. Conservation Tracking. Proceedings of the IEEE International Conference on Computer Vision (ICCV 2013), 2013.

Addressed in the combined review section.

4) I find that one of the main limitations to usability of this software is that installation seems hard. From the description in the web, I understand that I need all the following: (1) A Linux machine, (2) Python (and know what a virtual environment is and how to create it), (3) Matlab or Matlab's MCR, (4) git (and knowledge about how to clone a repository). Given that the target audience of this software are experimentalists in the field of animal behavior, I think that these requirements will severely damage its usability. And it seems that part of these requirements are in fact unnecessary (at least git should not be needed for basic users). And even having the necessary skills, the need of so many different steps makes me worry that something will fail along the way. I think that a modest investment in simplifying this step will increase the number of users substantially.

The somewhat complicated installation process is mostly due to the hybrid nature of our software, combining MATLAB and Python components. This does not allow us to use PyPI to directly install the software. We tried to compensate by providing a detailed step-by-step installation instruction that was tested with a few experimental ecologists. We are aware of this weak point, and plan to move to a pure Python implementation in the next version of anTraX, which will simplify the process tremendously. It will, however, take some time. For now, we followed the reviewer’s suggestion and added a direct download installation flow to the online documentation (https://antrax.readthedocs.io/en/latest/installation11/#Get-anTraX).